# Global evaluation of lineage-specific human papillomavirus capsid antigenicity using antibodies elicited by natural infection

Gathoni Kamuyu[1], Filomeno Coelho da Silva[1], Vanessa Tenet[2], John Schussler [3], Anna Godi[1], Rolando Herrero[4], Carolina Porras[4], Lisa Mirabello [5], John T. Schiller[6], Mónica S. Sierra [5], Aimée R. Kreimer [5], Gary M. Clifford [2] & Simon Beddows [1,7] ✉

Human Papillomavirus (HPV) type variants have been classified into lineages and sublineages based upon their whole genome sequence. Here we have examined the specificity of antibodies generated following natural infection with lineage variants of oncogenic types (HPV16, 18, 31, 33, 45, 52 and 58) by testing serum samples assembled from existing archives from women residing in Africa, The Americas, Asia or Europe against representative lineage-specific pseudoviruses for each genotype. We have subjected the resulting neutralizing antibody data to antigenic clustering methods and created relational antigenic profiles for each genotype to inform the delineation of lineage-specific serotypes. For most genotypes, there was evidence of differential recognition of lineage-specific antigens and in some cases of a sufficient magnitude to suggest that some lineages should be considered antigenically distinct within their respective genotypes. These data provide compelling evidence for a degree of lineage specificity within the humoral immune response following natural infection with oncogenic HPV.

Human papillomavirus (HPV) is the causative agent of cervical and other anogenital and head and neck cancers and accounts for >600,000 cancer cases globally per annum[1]. Around 40% of infection-related cancer cases and about 5% of all cancer cases worldwide, can be attributed to HPV.

HPVs exhibit a low evolutionary rate, typical of small dsDNA viruses. Nevertheless, many distinct genotypes have arisen over time[1]. A small number of genotypes from the Alpha-papillomavirus genus (HPV16, 18, 31, 33, 35, 39, 45, 51, 52, 56, 58, 59) contribute to the greatest disease burden, although the contribution of each genotype varies and there are some minor geographical biases in their distribution[2]. Whole

genome sequencing (WGS) has led to the identification and subsequent classification of sub-genotype lineage and sublineage variants as well as a better understanding of the underlying mechanisms driving HPV genome diversity[3,4]. In contrast to the similar global prevalence of carcinogenic HPV genotypes, lineage and sublineage variants exhibit markedly different phylogeographic distribution[5–10]. For example, HPV16 exists as distinct variant lineages (A–D) and sublineages (A1–4, B1–4, C1–4, and D1–4) which exhibit differential distribution across Africa, Asia, Europe, and The Americas[10]. There is growing evidence of an increased risk of disease associated with certain lineage variants for some genotypes[4,7–12]. Given the timeframe in

[1]Virus Reference Department, Public Health Microbiology Division, UK Health Security Agency, London, UK. [2]International Agency for Research on Cancer (IARC/WHO) Early Detection, Prevention and Infections Branch, Lyon, France. [3]Information Management Services Inc, Silver Spring, MD, USA. [4]Agencia Costarricense de Investigaciones Biomédicas (ACIB) formerly Proyecto Epidemiológico Guanacaste, Fundación INCIENSA (FUNIN), San José, Costa Rica. [5]Division of Cancer Epidemiology and Genetics, National Cancer Institute, Bethesda, MD, USA. [6]Center for Cancer Research, National Cancer Institute, Bethesda, MD, USA. [7]Blood Safety, Hepatitis, Sexually Transmitted Infections and HIV Division, UK Health Security Agency, London, UK. ✉e-mail: simon.beddows@ukhsa.gov.uk

which papillomaviruses have evolved in parallel with vertebrate species, attempts have also been made to relate HPV genome diversity with the emergence of the modern human species[13,14]. Thus, lineage and sublineage branches of the phylogeny are estimated to have emerged between 200–500 and 50–200 thousand years ago, respectively, around the time of the emergence of early modern humans. This is a relatively new area of research and there remain significant gaps in our understanding of the global diversity of HPV genotypic variants and in particular the impact of this variation on the structure and function of individual HPV proteins.

The HPV capsid consists of 360 copies of the major capsid protein, L1, arranged as 72 pentamers forming a $T = 7$ icosahedral lattice[15], with the minor capsid protein, L2, required for virus infectivity[16]. HPV pseudoviruses comprise L1/L2 capsids in which a plasmid expressing a marker protein is encapsidated. Surface-exposed external L1 loops are the target for neutralizing antibodies[17] and L1 virus-like particles form the basis of the licensed prophylactic vaccines[18]. Natural infection with HPV can persist for several months before being cleared by host cell-mediated immunity[1]. Although only a minority of individuals seroconvert following incident infection, and serum antibodies tend to be of low titer, they may nevertheless reduce the risk of new infections[19–23].

Capsid-based serology is an important tool for estimating vaccine-induced immunity as well as having utility for seroprevalence studies[19,24]. The antibody response against the L1 capsid is almost entirely type-specific, with some low-level cross-reactivity against related types, but the impact of antigenic variation on this type-specific antibody response is unclear[25–27]. Studies examining the potential impact of natural variation on capsid antigenicity have, in some cases, demonstrated variant-specific differences in sensitivity to

vaccine immune sera and monoclonal antibodies (MAbs)[28–36], but there are few studies examining the potential impact of capsid variation on recognition by natural infection derived antibodies and these are limited to a small number of sera and variants[30,31,36]. This differential antigenicity may render some lineages less sensitive to antibodies elicited following natural infection with another more genetically distant lineage leading to the possible identification of diagnostic antigenic motifs, the establishment of lineage-specific serotypes for some genotypes, and insights into the evolution of HPVs into hundreds of distinct genotypes/serotypes.

For this study, we assembled a large panel of serum collected from women resident in Africa, The Americas, Asia, and Europe following natural infection with lineage-specific variants of vaccine-preventable types (HPV16, 18, 31, 33, 45, 52, and 58). We generated functional antigenicity data using lineage-specific pseudovirus (PsV)-based neutralization assays in which the resulting neutralizing antibody data were subjected to hierarchical clustering and antigenic cartography to create relational antigenic profiles for each genotype. These data delineate the lineage-specific antigenic relationship between capsid proteins of several oncogenic HPV genotypes and improve our understanding of lineage-specific immunity in HPV natural infection.

## Results

Serum (or plasma) samples were collected from women with assigned lineage-specific infection and evaluated in a neutralization assay using representative lineage-specific pseudoviruses (PsV) for each genotype (HPV16, 18, 31, 33, 45, 52 and 58) under study (Fig. 1). The resulting data were subjected to hierarchical clustering and antigenic cartography to determine whether there was any evidence of lineage-specific antigenicity for that genotype.

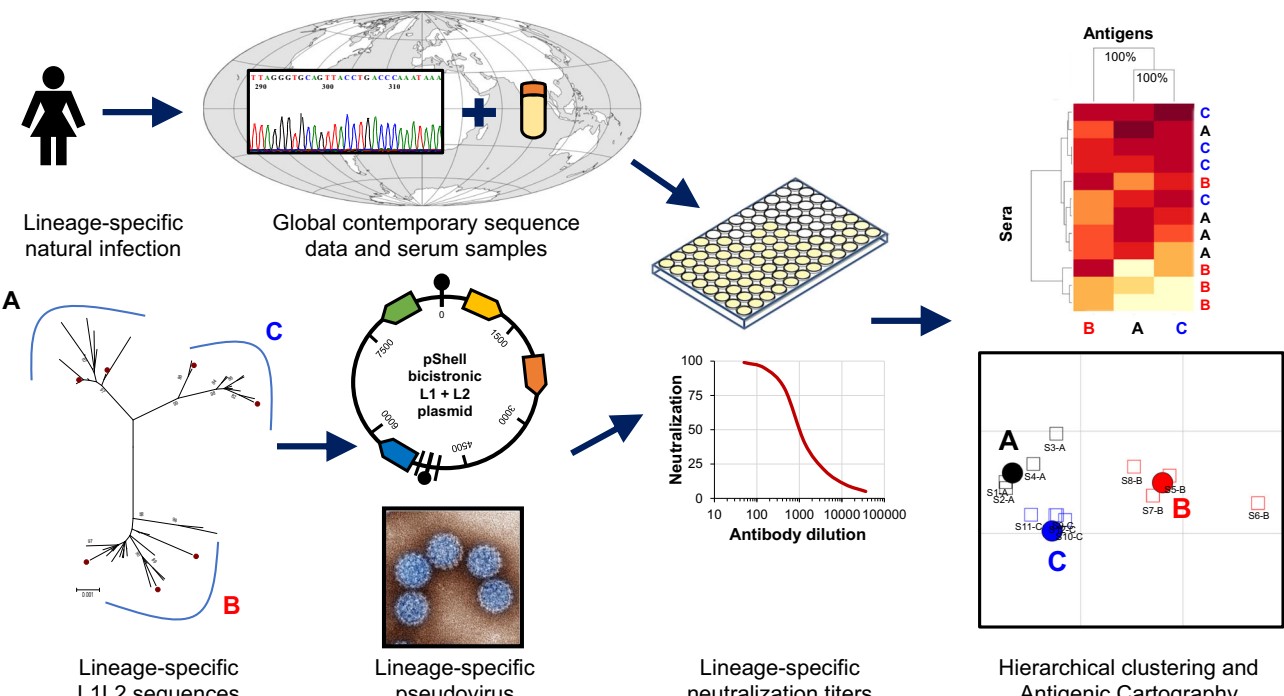

**Fig. 1 | Schematic depicting study execution.** Residual archived cervical samples from self-reported females collected for natural history or baseline vaccine studies that had previously been genotyped were assigned a specific lineage following whole genome or Sanger sequencing. Where available, contemporaneously collected serum (or plasma) samples representing defined lineages for each of the following genotypes (HPV16, 18, 31, 33, 45, 52, and 58) were assembled. Lineage-specific L1 and L2 sequences were downloaded from the NCBI database and bicistronic psheLL vectors containing the majority of L1 and L2 inserts together with a luciferase reporter plasmid was used to create lineage-specific PsVs following transfection of 293TT cells. Lineage-specific serum (or plasma) and lineage-specific PsV were used in neutralization assays to generate lineage-specific neutralization titers. Neutralizing antibody data were subjected to hierarchical clustering and antigenic cartography methods to create relational antigenic profiles for each genotype. World map courtesy of Tentotwo (https://commons.wikimedia.org/w/index.php?curid=21507855).

## Sample collection and seropositivity rates

A panel of 2255 serum (or plasma) samples representing lineage-specific natural infections was assembled (Supplementary Fig. 1). The collection was not equally representative of each continent: The Americas (1270; 56%), Africa (575; 25%), Asia (373; 17%), Europe (37; 2%), and Oceania (0; 0%). The majority (2208/2255; 98%) of serum (or plasma) samples were collected at the same time as the sample collected for HPV DNA analysis (to determine genotype and lineage), with the majority (44/47; 94%) of the remainder collected within 2 years following DNA sample collection.

An a priori target of 150 samples per lineage was met (or nearly met, i.e., within 10%) for 12 of the 23 (52%) lineages examined, with the distribution of samples skewed in favor of the most common lineages. The mean seropositivity rate (754/2255; 33%, 95% CI 31–35%) was higher than the anticipated rate of 10–20% (Supplementary Table 1). Seropositivity rates were largely unaffected by sample type (except HPV18, 45), disease status (except HPV18, 45), geographic region (except HPV16), and lineage assignment. The target of 15 seropositive samples per lineage was met for 70% (16/23) of lineages.

## Lineage-specific neutralization titers

There were several instances of lineage-specific neutralizing antibody reactivity (Fig. 2, Supplementary Table 2). HPV16 lineage B sera were less reactive against the HPV16 D antigen and HPV16 C sera preferentially recognized the HPV16 C antigen. Both HPV18 lineage A and B sera were less reactive against the HPV18 C antigen; there were few HPV18 lineage C sera in the study, and all were seronegative preventing an assessment of the reciprocal response (Supplementary Table 1). HPV31 lineage A sera were less reactive against the HPV31 C antigen, while lineage C sera preferentially reacted with the HPV31 C antigen. HPV33 lineage A sera were less reactive against both the HPV33 B and C antigens, while lineage B sera preferentially recognized the HPV33 B antigen. HPV45 lineage A sera were less reactive against the HPV45 B antigen, but lineage B sera were equally reactive against A and B antigens. HPV52 lineage A sera preferentially recognized the HPV52 A antigen while both lineage B and C sera recognized lineages A and C antigens in preference to the HPV52 B and D antigens. HPV58 lineage A sera poorly recognized the HPV58 C antigen. In some cases, a tendency towards a preference could be observed, e.g., HPV52 lineage D sera reacted against HPV52 D antigen seemingly in preference to lineages A–C but anecdotal observations without statistical support should be interpreted with caution.

These differential neutralizing antibody titers manifest as reduced seropositivity against the reference lineage A antigen in some cases (Supplementary Table 2). For example, of 64 sera positive for neutralizing antibodies against outlier lineage antigen HPV16 C only 50 (78%; 95% CI, 66–85%) were positive against HPV16 lineage A.

## Two-dimensional antigenic clustering

Natural log-transformed neutralizing antibody titer data were subjected to two-dimensional hierarchical clustering. Data were reordered according to serum and antigen dendrograms constructed from the resulting distance matrices to generate a more granular view of the serum: antigen interactions within a genotype (Fig. 3). Lineage-specific antigens were distinguishable with a high level of bootstrap support (100% of 500 iterations) with the antigenic dendrograms also supporting the delineation of distinct antigenic clusters in most cases. HPV16 lineages A and B clustered together while HPV16 lineage C was an outlier antigen. HPV31 lineages A and B clustered together while lineage C represented an outlier antigen. For HPV33 lineages B and C constituted a distinct group separate from lineage A, while for HPV52 and HPV58 lineages D and C represented outlier lineages, respectively. To corroborate the apparent segregation of lineage antigens identified here we made use of an additional hierarchical clustering tool (Supplementary Fig. 2). Taken together, these data suggest that the amino

acid polymorphisms indicative of distinct lineages bestow differential antigenic properties on the HPV capsid proteins and that this phenomenon was evident to some degree for all the genotypes tested.

Hierarchical clustering is a useful tool for delineating distinct antigen clusters, but it is quite sensitive to small consistent differences within a dataset and gives no indication of antigenic distance. Antigenic cartography was therefore employed to map the relative distance between lineage-specific antigens (Fig. 4). For this purpose, we considered a ≥ 2-fold distance (1 antigenic unit) between two antigens to be significant, but our a priori threshold for antigens to be considered serologically distinct was 4-fold (2 antigenic units). HPV16 lineages A and B were indistinguishable antigenically (<2-fold), while lineage C antigen exhibited an estimated distance of 2.7-fold from lineage A antigen. The mean estimated distance following 10 resampling iterations using 90% of the data was 2.7 (95% CI 2.6–2.7), demonstrating that this is a consistent and robust estimate (Supplementary Fig. 3). These data corroborate the hierarchical clustering analysis and suggest that the HPV16 lineage C antigen is significantly distant from the other lineage antigens within this genotype. For HPV18 and HPV31 all antigens were resolved within a 2-fold distance of each other, suggesting that despite a degree of separation by hierarchical clustering these antigens should be considered antigenically indistinguishable. HPV33 lineage B and C antigens were estimated to be 4.2- and 5.9-fold distance from lineage A with mean estimates from the 90% resampling iterations of 4.2 (95% CI 4.1–4.3) and 5.9 (5.7–6.1), respectively, and should be considered antigenically distinct from lineage A. With only two lineages representing HPV45, we were unable to generate an antigenic map. To address this shortcoming, we performed two antigenic map simulations using an additional copy of the lineage A (designated A′) or B (designated B′) dataset resulting in an antigenic distance estimate of 1.8-fold for both A′ and B′ simulations (Supplementary Fig. 4). HPV52 lineages A and C were antigenically indistinguishable (<2-fold distance) with antigens B (2.3-fold) and D (3.9-fold) being significantly distant to lineage A, although not meeting the criterion to be classified as antigenically distinct. HPV58 lineage C antigen was a clear outlier, being 17.2-fold distant (mean 17.0; 95% CI 16.1–18.0 from ten resampling iterations) from lineage A antigen and should be considered antigenically distinct, in contrast to lineages B and D which clustered within 2-fold of lineage A. To corroborate the relative positions of the lineage antigens in two-dimensional space we also made use of principal component analysis (Supplementary Fig. 2). Taken together, these assessments suggest that the major capsid proteins of some lineages exhibit differential antigenicity within their respective genotype of a magnitude between 2 and 20-fold from that of the reference lineage A antigen. For comparison purposes, antigenic maps were created using published type-specific neutralizing antibody data generated using rabbit antisera wherein the antigenic distance estimates between any two types were estimated to be ≥100 fold (Supplementary Fig. 5)[25].

A small proportion of individuals included in this study had evidence in the accompanying DNA sample of dual (192/2255; 8.5%) or multiple (18/2255; 0.8%) genotype infections. To evaluate any potential confounding of the antigenic distance estimates by inclusion of these individuals, we generated estimates of the antigenic distances between lineages for each genotype including only those individuals harboring single infections (Supplementary Fig. 6). The resulting distances were essentially the same as for the complete dataset, suggesting little or no confounding of the antigenic distance estimates in this study using samples from individuals with multiple infections.

We conducted further impact analyses to test the robustness of the antigenic distance estimates. In the first instance, we evaluated the potential impact of the geographical origin of the serum samples and an artificially reduced seropositivity rate. We used the HPV16 dataset as it was well represented by samples from The Americas,

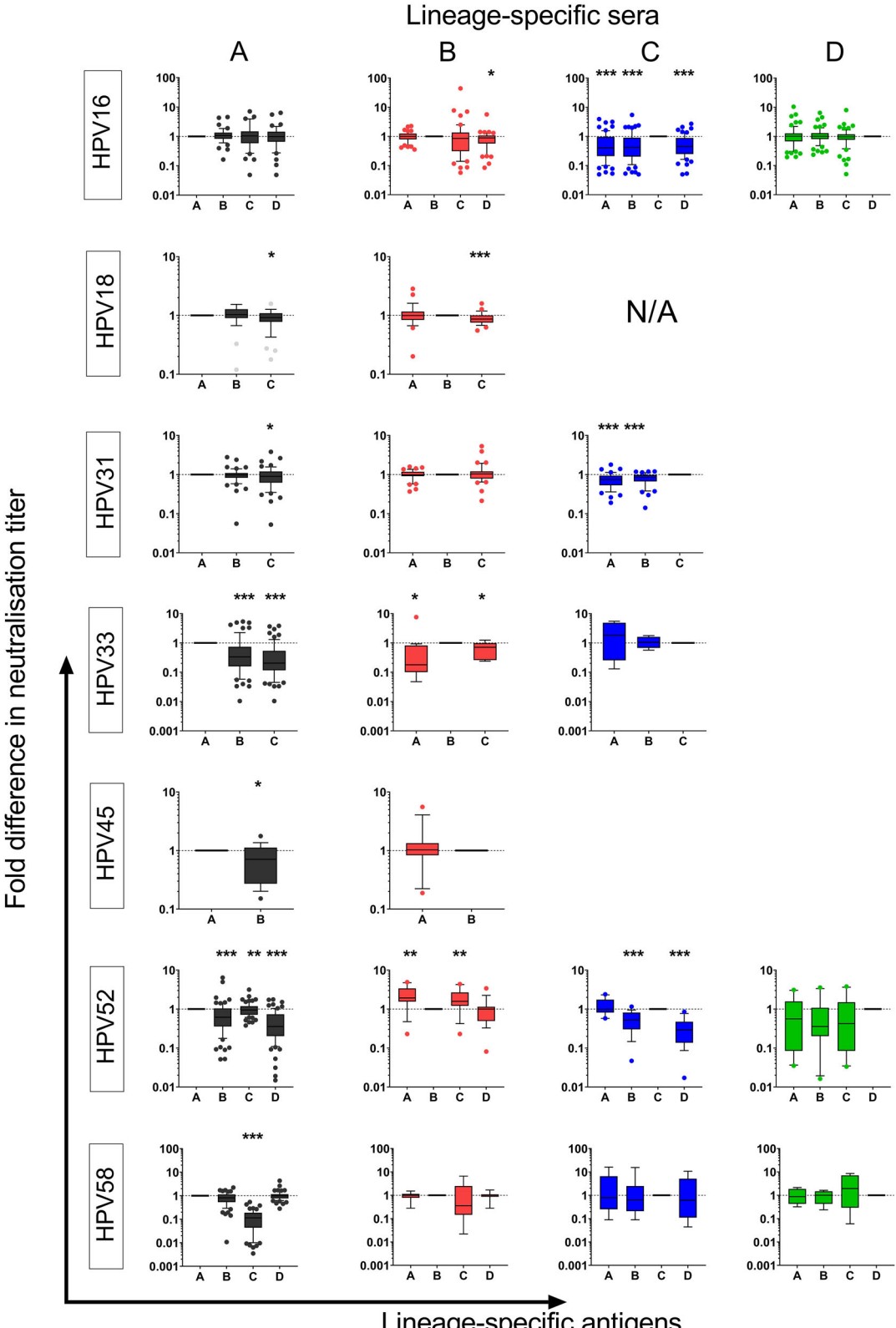

**Fig. 2 | Lineage-specific neutralizing antibody titers.** Serum samples derived from lineage-specific infections (A, black; B, red; C, blue and D, green) evaluated against PsV antigens for each genotype. The *n* sample number is variable for each plot and is stated in Supplementary Table 2. For example, the number of sera representing HPV16 lineages A (*n* = 46), B (*n* = 50), C (*n* = 64), and D (*n* = 66) are indicated in parentheses. Box (median, IQR) and whisker (10th and 90th percentiles; filled circles represent outliers) plots of the ratio of the neutralization titer against each PsV antigen over that of the PsV representing the assigned lineage-specific infection for each assessment. Pairwise comparisons made using Wilcoxon paired signed rank test (two-sided, exact *p* values) where *\*p* < 0.05 (HPV16 B *p* = 0.021; HPV18 A *p* = 0.043; HPV31 A *p* = 0.024; HPV33 B *p* = 0.017 and *p* = 0.032; HPV45 A *p* = 0.040); *\*\*p* < 0.01 (HPV52 A *p* = 0.006; HPV52 B *p* = 0.004 and *p* = 0.008); or *\*\*\*p* ≤ 0.001. N/A, not applicable.

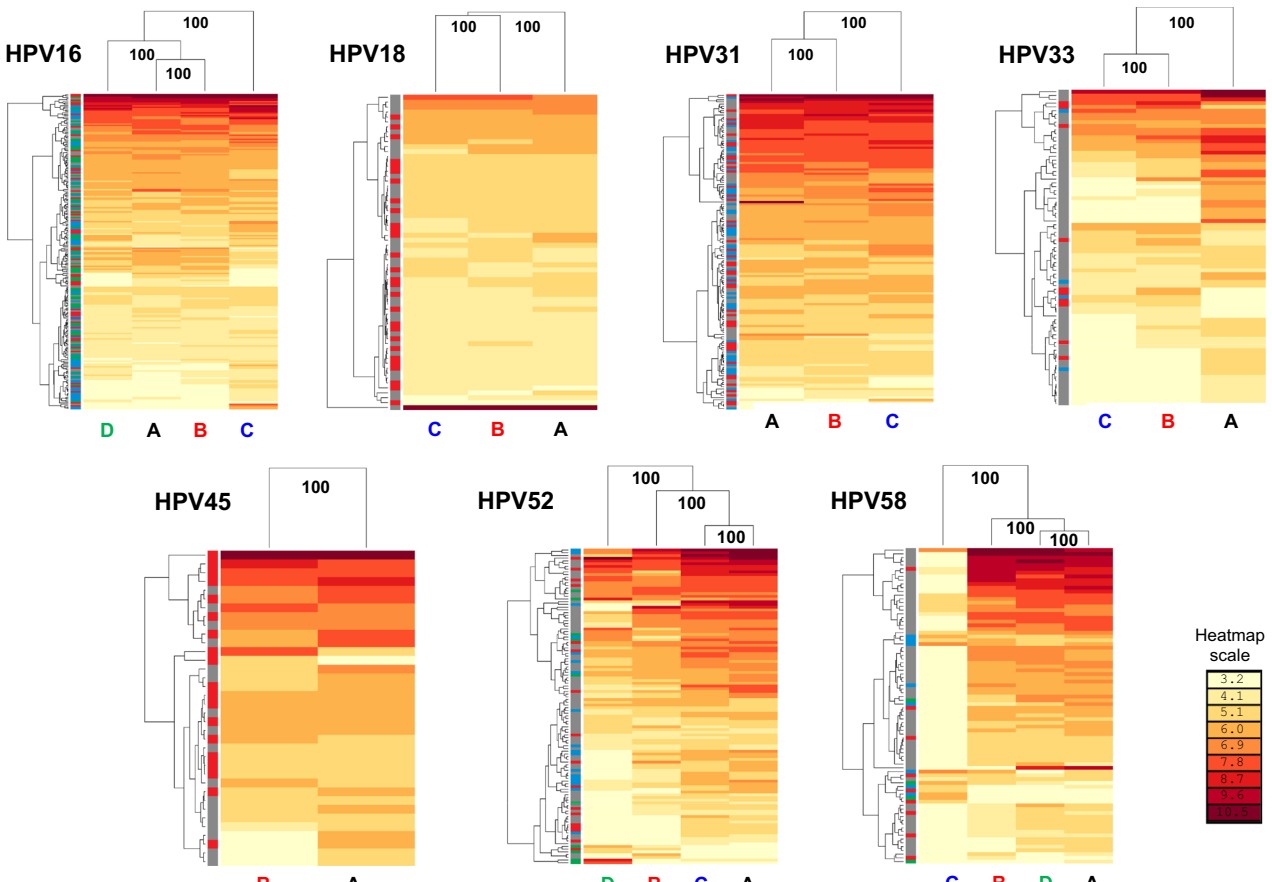

**Fig. 3 | Hierarchical clustering of neutralizing antibody titers.** Neutralizing antibody titers were natural log-transformed and subjected to two-dimensional hierarchical clustering with resulting data re-ordered according to serological (left) and pseudovirus antigen (top) dendrograms constructed from the resulting Euclidean distance matrices with the antigen clusters supported by the indicated percentage of 500 bootstrapped pseudoreplicates. Lineage-specific PsV antigens labeled at the base of the heatmap (A, black; B, red; C, blue; D, green). Serum sidebar denotes the natural infection lineage from which the serum (or plasma) sample is derived: A (gray), B (red), C (blue), and D (green). The antigen clusters are supported by data from many sera whereas the serum clusters are supported by data from relatively few antigens; thus, more weight should be given to the definition of antigen clusters that have bootstrap support. The indicative natural log heatmap scale bar is shown.

Africa, and Asia and therefore amenable to such analyses (Supplementary Fig. 7). The antigenic distance estimate between lineages A and C, for example, was impacted slightly by the geographical source of the samples included in the analyses, particularly by samples from Africa. It should be noted, however, that the sample sizes for some of these evaluations were quite low which could affect the outcome by including fewer samples representing a particular lineage, such as a reduced number of lineage C sera from The Americas and Asia. Geographical stratification was not reliable for other genotypes (for example HPV52 and HPV58) due to the lack of lineage-specific seropositive samples for some geographical regions. A 50% reduction in the number of seropositive samples representing each lineage had little impact on the antigenic distance estimates for this type. We also attempted to compensate for the low number of samples representing some of the outlier lineages of HPV33, HPV52, and HPV58 by conducting two resampling with replacement evaluations which created pseudo datasets with sample sizes closer to the expected 15–30 seropositive samples per lineage. The outlier lineages identified following these evaluations were the same as those identified for the original dataset, although there were some minor differences in the distance estimates depending on whether the structure of the original dataset was maintained (Supplementary Fig. 8). Overall, these impact analyses support the robustness of the antigenic distance estimates.

## Chimeric and mutant pseudoviruses

A more detailed comparison was made between the antigenic profiles of the lineage A (reference) antigen and an outlier lineage antigen for three genotypes that had demonstrated significant lineage-specific antigenic distances: HPV33 (lineage B), HPV52 (lineage D) and HPV58 (lineage C) (Fig. 5). Neutralizing antibody data using chimeric PsV, wherein the L1 and L2 gene inserts were swapped between the reference and outlier lineages (e.g., HPV33 $A_{L1}/B_{L2}$ and HPV33 $B_{L1}/A_{L2}$) support the concept that the majority, if not the entirety, of neutralizing antibodies generated during natural infection target the major capsid protein, with little or none of the antibody repertoire raised against the minor capsid protein. Lineage-specific L1 surface-exposed motifs were swapped between relevant antigens to assess the contribution of specific amino acid residues. For example, HPV52 $A_{FGHI}$ is a lineage A PsV with the FGHI residues of HPV52 lineage D (K281, T354T, D357) while HPV52 $D_{FGHI}$ is a lineage D PsV with the FGHI residues of HPV52 lineage A (Q281, K354, S357). HPV52 lineage A sera preferentially recognized PsV HPV52 $D_{FGHI}$ while lineage D sera preferentially recognized PsV HPV52 $A_{FGHI}$ PsV. Taken together, these data suggest that the differences in neutralizing antibody specificity observed here between the reference A lineage and the outlier lineage for genotypes HPV33, HPV52, and HPV58 are due to a limited number of amino acid residues found within the surface exposed loops of the major capsid protein.

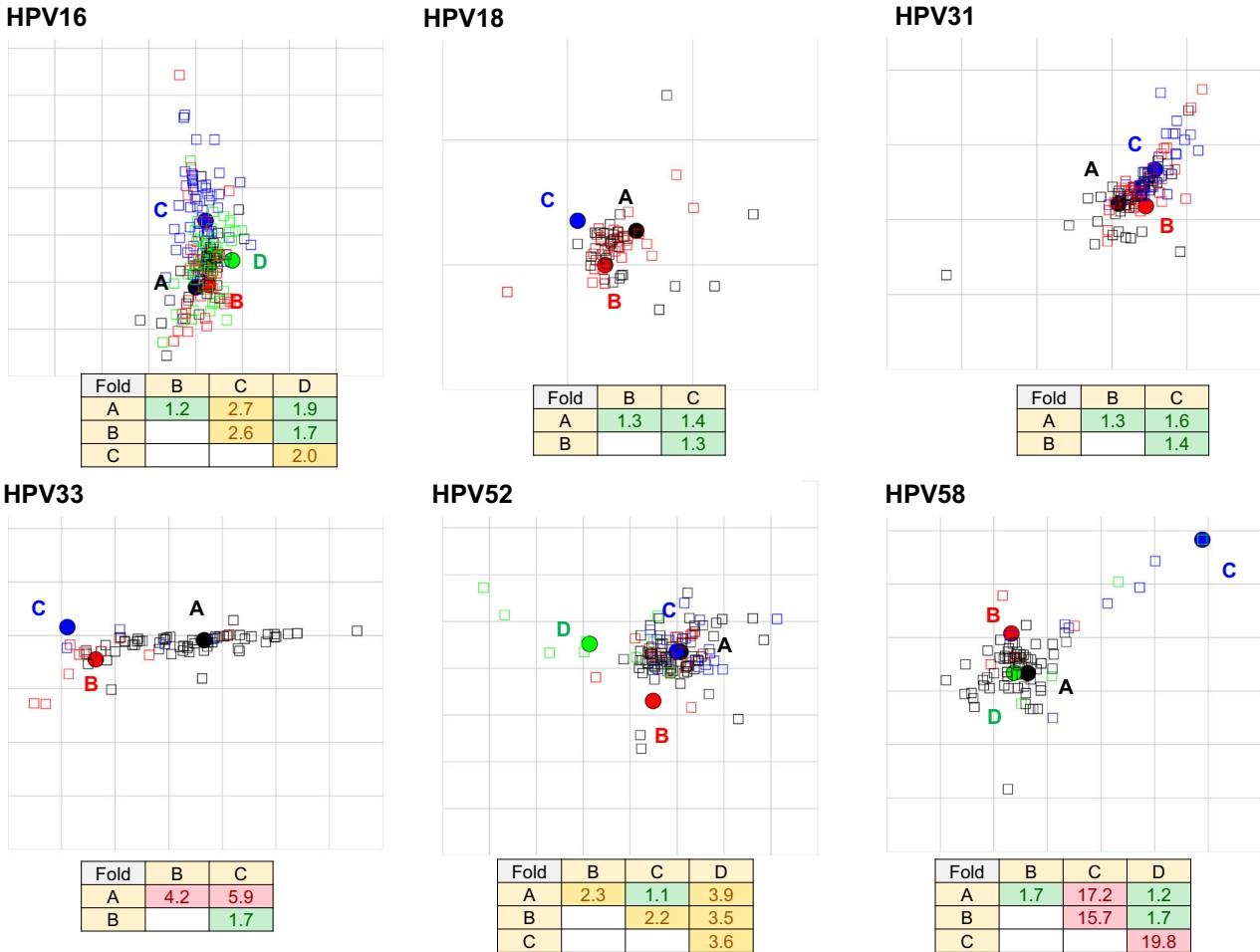

**Fig. 4 | Lineage-specific antigenic cartography.** Neutralizing antibody titers were subjected to two-dimensional clustering using the antigenic cartography algorithm. Filled in circles and open squares represent lineage A (black), B (red), C (blue), and D (green) PsV antigens and serum (or plasma), respectively. In each antigenic map regardless of actual size, the gray grid squares represent 1 antigenic unit (AU), which is equivalent to a 2-fold distance between antigens; thus, three grid squares are equivalent to an 8-fold ($2^3$) distance. While individual serum samples are derived from specific lineage infections and expected to coalesce around their respective antigen, there is no expectation of a strict lineage antibody specificity as sera will likely display a range of antibody specificities and may in some cases be mapped closer to a heterologous antigen than the corresponding homologous antigen. Antigenic distance estimates are derived from the sum of the responses of all the sera tested and are the primary outcome measure from these evaluations. Fold difference between indicated PsV antigens shown with <2-fold (green), 2–4 fold (gold), and >4-fold (red) differences highlighted as indicated.

## Discussion

We assembled a large panel of serum samples collected from women resident in Africa, The Americas, Asia, and Europe with known lineage-specific natural infection with an oncogenic HPV genotype and conducted a comprehensive evaluation of the lineage specificity of the neutralizing antibodies elicited. The resulting functional antibody data were then subjected to antigenic clustering methods in order to create relational antigenic profiles for each genotype and inform the delineation of lineage-specific serotypes, improve our understanding of lineage-specific immunity in HPV natural infection, and assess implications for serosurveillance studies using reference antigens.

In all cases, hierarchical clustering highlighted differential serological recognition of the lineage-specific antigens within each genotype. These data suggest that the extent of lineage diversity within a genotype is sufficient to confer differential antigenic properties on the HPV structural proteins for all the genotypes tested. Hierarchical clustering is a useful tool for assessing pattern recognition within a large immunological dataset[37], but it is sensitive to small, consistent changes and does not give any indication of scale. We therefore made use of antigenic cartography to estimate the antigenic distance between lineage-specific antigens for each genotype. In addition to the

Influenza virus, antigenic cartography has been used to map the antigenic profiles of a range of pathogens including Dengue virus, Zika virus, Norovirus, and more recently SARS-CoV-2[38–41]. We set an a priori threshold of 2 antigenic units or a 4-fold distance before we considered two lineages as being antigenically distinct. HPV18, HPV31, and HPV45 demonstrated antigenic distances between their lineage-specific antigens of <1 antigenic unit (or <2 fold), and thus for these genotypes their lineages should be considered to belong to the same serotype. For HPV16 and HPV52, we found notable and consistent antigenic distances between lineage-specific antigens (between 1 and 2 antigenic units; 2-4-fold) although as these estimates were below the 4-fold threshold we consider that the lineages of these genotypes belong to the same serotype. For HPV33 and HPV58 significant antigenic distances were found between lineage-specific antigens (>2 antigenic units; >4 fold) and so the lineages of these genotypes, particularly HPV33 B/C and HPV58C should be considered as serologically distinct from the reference lineage A antigen, according to our a priori threshold. The estimated antigenic distances were robust and reproducible following removal of individuals with evidence of mixed infection suggesting that mixed infection samples were too few to influence the antigenic distance estimates or that there is no intrinsic

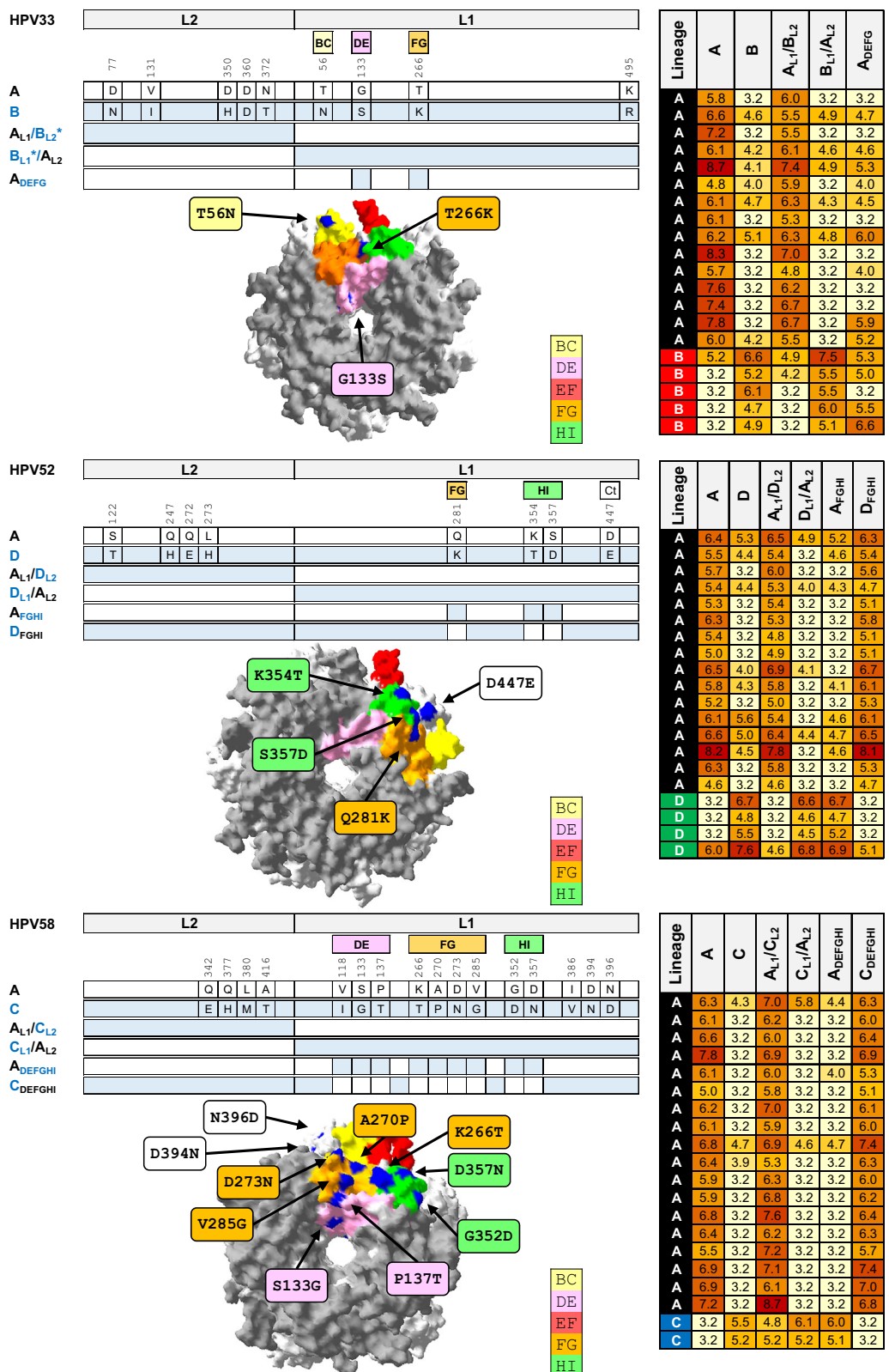

**Fig. 5 | Fine mapping of HPV capsid antigenicity.** Mutational analysis of reference (lineage A) and outlier lineage for HPV33 (top panel; lineage B), HPV52 (middle panel; lineage D), and HPV58 (bottom panel; lineage C). Each panel comprises a cartoon depicting chimeric and mutant PsV sequences from reference lineage A and outlier lineage for each genotype. Heatmap presents natural log-transformed neutralizing antibody titers against each antigen using sera representing reference and outlier lineages. Capsomer crystal images representing HPV33 (PDB accession number: 6IGE.2), HPV52 (PDB accession number: 6IGF.1), and HPV58 (PDB accession number: 5Y9E.1) highlight variable residues between indicated lineages.

Surface-exposed loops are highlighted as indicated in the key: BC (yellow), DE (pink), EF (red), FG (gold), and HI (green). Site polymorphisms were reported using standard nomenclature as X123Y wherein the first residue denotes the amino acid residue in the reference sequence, its position in that sequence, and then the residue in the outlier sequence at that same position. *, L2 fragment in clone $A_{L1}/B_{L2}$ differs from lineage B at two residues (V131 and N360), but shares three similar polymorphisms (N77, H350 and T372) and is otherwise identical to lineage B L2; L1 fragment in clone $B_{L1}/A_{L2}$ differs from lineage B by one residue (K495) but is otherwise identical to lineage B L1.

capability for mixed infection samples to interfere with the outcome. These lineage-specific antigenic distance estimates were of a relatively minor magnitude compared to estimates of type-specific antigenic distance using pre-clinical immune sera.

We also conducted an in-depth mutational analysis using PsV representing an outlier lineage and the reference lineage for genotypes HPV33, HPV52, and HPV58. These data demonstrated that the majority of antibodies elicited during natural infection, at least for the lineages evaluated here, target small antigenic domains on the surface of the major capsid protein wherein lineage-specific residue polymorphisms bestow differential recognition by lineage-specific natural infection antibodies. The antigenically distinct lineages defined by natural infection sera in this study were the same lineages found to be antigenically distinguishable when using nonavalent vaccine sera[35] or MAbs[33], which corroborate the findings in this report including, where evaluated, reciprocal lineage-specific responses using murine antisera[31,36]. Furthermore, the mutational analysis conducted within this study identified specific antigenic motifs on the surface of the major capsid protein that were similar to those motifs identified using MAbs and murine antisera suggesting some degree of overlap in specificity between these antibody repertoires[31,33,36].

The lineage diversity within the L1 and L2 open-reading frames (ORFs) is a fraction of the whole genome variation between lineages within a genotype that has evolved over millennia[13,14]. There is mounting evidence to support certain lineages within some genotypes being associated with worse disease outcomes[7–12] although the mechanism is uncertain[42]. The impact of lineage variation on the structure and/or function of other HPV ORFs or regulatory elements is unclear due to the limited number of studies conducted and further research in this area may help to elucidate the underlying mechanisms that contribute to differential disease outcomes.

An analysis of the distribution of synonymous and non-synonymous sites within the L1 and L2 ORFs suggests that overall these genes are under purifying selection[4], most likely to maintain the structural and functional integrity of their respective gene products, although a mixture of positive and purifying selection pressures across individual sites across the ORFs cannot be ruled out. The lineage-specific neutralizing antibody responses identified here, particularly for those that are reciprocal in nature, may suggest that at least some of the residues on the surface of the major capsid protein have arisen and/or have been maintained due to humoral immune selection, although random drift in historically isolated populations may also have played a role.

There are several shortcomings associated with this study. While the scope of this study was substantial and the number of samples included significant, the samples were not evenly distributed across the genotypes, or lineages within each genotype, and so the a priori target of 150 samples per lineage was only met for just over half of the lineages examined. In addition, sample collection was skewed in favor of the most common lineages, as one might expect, such that samples representing lineage A met the a priori target whilst other lineages were represented by fewer samples. This would necessarily introduce a potential bias whereby the inter-antigen distances would be more likely to be influenced by the relationship between lineage A sera and the other antigens and less so by the reciprocal specificity. However, this target sample size was estimated based upon a 10–20% seropositivity rate[20,22] so that the resulting antigenic clustering analyses would include our target of 15 positive samples. In practice, seropositivity rates for lineages within a genotype were similar and averaged more than 30% which helped to compensate for the limited number of available samples for some lineages. In all cases, apart from HPV18 C (which is a rare lineage found in Africa), each lineage for each genotype was represented by at least some seropositive samples with 70% of lineages represented by 15 or more seropositive samples. Samples from Africa, The Americas, and Asia were well represented but

samples from Europe were lacking which could have introduced a bias in terms of the underrepresentation of lineages only to be found in Europe, but this was not the case; most lineages that were underrepresented were rare lineages from sub-Saharan Africa. The reciprocal nature of the antigen specificity was not always evident from the summary statistics and the precision around these estimates was further hampered by the fewer number of samples from the most distant lineages. Nevertheless, the mutational analysis provided supportive evidence for the reciprocity of this antigen specificity, at least for the antigens evaluated. To mitigate against some of these potential shortcomings, we collected a significant amount of quality assurance data to demonstrate the reproducibility of the testing. In addition, we performed two hierarchical clustering methods, principal component analysis, and antigenic cartography on the seropositive samples to corroborate any perceived specificity differences between the antigens. Furthermore, we conducted several sensitivity analyses to test the robustness of these estimates including iterations of the antigenic cartography wherein 90% of data were randomly selected (without replacement) for each type, the analysis of samples representing specific geographic regions, and simulating a reduced lineage-specific seropositivity rate for HPV16 as well as conducting proportionate and disproportionate random resampling (with replacement) evaluations for HPV33, HPV52 and HPV58. In all cases, these evaluations generated similar antigenic distance estimates to that of the original dataset suggesting that the antigenic distance estimates were robust, although the influence of the geographical origin of the samples on these estimates cannot be ruled out. For example, the antigenic distance between HPV16 lineages A and C approached 3-fold for African samples and 2-fold for samples from Asia and Europe only, suggesting some influence of the geographical origin of the samples on these estimates. Similar analyses for other genotypes were not possible due to the low number of lineage-specific seropositive samples in some geographical regions. We also cannot rule out the possibility that the antibody response being evaluated in some individuals is the result of a previous infection by another lineage of the same genotype not detected at the time of DNA testing. The number of co-infections with another genotype was rare for the individuals within this study (<10%) and the majority (98%) of samples were collected contemporaneous to a documented lineage-specific infection which together with the differential geographical dispersal of these lineages suggests that such an event would be extremely rare and should not be considered a significant confounding factor. Finally, although HPV PsVs have been used widely to monitor antibody responses to vaccines and natural infection, as well as elucidate steps in the entry process, differences between how PsVs behave in vitro and how authentic HPVs behave in vivo are uncertain, which is a limitation of all PsV-based systems.

This study provides compelling evidence for a degree of lineage specificity within the humoral immune response following natural infection with oncogenic HPV. These data have implications for HPV seroprevalence studies, for example, where antigens based upon the reference sequence may not be recognized by a significant proportion of infections in that country or region. This may lead to an underestimate of HPV exposure metrics for natural history studies or for baseline prevalence estimates prior to the implementation of national nonavalent vaccine programmes. Thus, for some studies, it may be worth considering using multiple lineage-specific antigens to fully represent the diversity within that region. These data further inform our understanding of the specificity of antibodies elicited following natural infection compared to those generated following vaccination. There is growing evidence that the detection of antibodies following natural infection with an HPV genotype can afford some protection against reinfection with that same genotype[19,43–45]. Whether antibodies are the primary effector in this setting, similar to the response following vaccination[46], or whether antibodies act in part as a proxy for a broader response that includes cell-mediated immunity is not clear.

However, if antibodies do play a significant role in mediating this protection, then it would be reasonable to expect that the antibody-mediated lineage specificity highlighted in this study may reduce the level of protection against infection by another more divergent lineage within the same genotype, subject to the individual lineage-specific prevalence. These data also support some degree of overlap between the functional antibody response elicited by natural infection and that generated by vaccination and following the immunization of small animals with the major capsid protein[31,35,36]. The ecological diversity of HPV subtypes and their variants within a country or region might be expected to change following the introduction of national vaccine programmes, as has been suggested for non-vaccine targeted genotypes[47]. It should be noted, however, that there is little evidence to support differential vaccine efficacy against HPV variants[48] and that vaccine-induced immunity is at least an order of magnitude greater than that found in natural infection[46]. In addition, in vitro assays may underestimate the level of protective antibodies compared to in vivo models[49].

While the antigenic motifs identified in this and previous studies[30–36] differ between and within genotypes, there are nevertheless some common residues implicated, suggesting a degree of shared topography for the capsid proteins of oncogenic HPV genotypes. These data imply that specific non-synonymous mutations found within the HPV capsid genes that have emerged over millennia along with the rest of the HPV genome directly affect the antigenicity of the proteins that they encode.

In summary, these data help to delineate the lineage-specific antigenic relationship between capsid proteins of several oncogenic HPV genotypes and improve our understanding of lineage-specific immunity in HPV natural infection.

## Methods
### Samples
Blood (serum or plasma) samples were made available for this study from existing archives held at The International Agency for Research on Cancer (IARC, Lyon, France) and the National Cancer Institute (NCI, Bethesda, USA) under study-specific material transfer agreements. Individual female (self-reported) participants were identified for whom a confirmed current or prior (<2 years from blood sample collection) lineage-specific infection was recorded and for whom an accompanying blood sample was available. Samples were originally collected from ethically approved studies[50–59] and ethical approval for the use of these samples for the current study was made following approval from local institutional (IARC, NCI) review boards.

The IARC has coordinated international surveys of cervical cancer specimens[50,51], a multicenter case-control study[50], and population-based HPV prevalence surveys[52–56] in many countries, that collected blood samples from women with paired cervical samples of known HPV genotype. Samples were all collected between 1989 and 2004, prior to the implementation of HPV vaccination programmes. All protocols were approved by the IARC (https://ethics.iarc.fr/) and local ethics committees, and informed, written consent was obtained from all participants. Cervical samples (exfoliated cells or tissue/biopsy specimens) were genotyped for a broad range of HPV types using broad spectrum PCR-based assays, using MY09/11 and/or GP5+/6+ consensus probes. Extracted DNA from HPV16/18/31/33/45/52/58 positive cervical samples in the IARC biobank were sent to the NCI, National Institutes of Health, Rockville, MD (NCI) for whole HPV viral genome sequencing and classification by viral (sub)lineage[10,60].

The NCI contributed serum samples from participants in the control arms of the Costa Rica HPV Vaccine Trial (CVT, Clinicaltrials.gov NCT00128661) and plasma samples from participants in the Guanacaste Natural History Study (NHS)[57–59]. All study protocols were approved by the U.S. National Cancer Institute (NCI) Institutional Review Boards and the corresponding Costa Rican Institutional Review Board (CEC-CENDEISS, CEC-INCIENSA, CEC-UNA, CEC-ICIC); all participants signed written informed consent.

Blood samples were available from the indicated countries representing the following regions: Africa (Algeria, Guinea, Kenya, Mali, Morocco, Nigeria, Tanzania, Uganda), The Americas (Argentina, Bolivia, Brazil, Chile, Colombia, Costa Rica, Cuba, Panama, Paraguay, Peru), Asia (India, Indonesia, Korea, Philippines, Thailand, Vietnam), and Europe (Poland, Spain) (Supplementary Fig. 1). Although samples were obtained from women resident in these geographical regions at the time of collection no inference is made about the genetic ancestry or ethnicity of these women on this basis.

The testing laboratory received coded serum (or plasma) samples representing the infecting genotype for that individual but was blinded to the associated metadata including assignment of natural infection lineage, sample type (serum or plasma), disease state (cervical cancer case versus cancer-free controls) and geographical origin (region, country) until testing was complete. Additional coded samples were included to represent the small number of mixed infections present in the cohort.

### Pseudovirus neutralization assay
The expression, purification, and use of L1L2 PsV variants of HPV16, 18, 31, 33, 45, 52, and 58 have been described previously[35]. The specific PsV clones that were used in this study to represent each lineage (A–D) were based upon the majority representative consensus lineage or sublineage sequence as follows: HPV16 (A–D), HPV18 (A–C), HPV31 (A1 reference, B2, C), HPV33 (A1, B, C), HPV45 (A2, B2), HPV52 (A1, B2, C, D) and HPV58 (A2, B2, C, D1) for which the details can be found in the Supplementary Fig. to the aforementioned publication[35]. PsVs were expressed in 293TT cells (https://www.atcc.org/products/crl-3467), purified by ultracentrifugation, and quality assessed by SDS–PAGE, negative stain electron microscopy, and infectivity[61]. Serum or plasma samples were screened against PsVs representing each lineage for that genotype and sera positive for neutralizing antibodies against any lineage were then titrated against all lineages representing that genotype[61]. Each sample was titrated once for the primary dataset but a subset of seropositive samples (mean 4%; range 1–7% of total samples depending on genotype; $n = 101$ total samples) were subjected to repeat testing for quality assurance purposes, resulting in a median $\log_{10}$ titer ratio of 0.98 (inter-quartile range, IQR, 0.94–1.01) and a Pearson's correlation coefficient ($r^2$) of 0.923 for the initial and repeat tests. In some cases, indicated PsV mutants representing HPV33[32], HPV52[31], and HPV58[36] genotypes were used to evaluate the specificity of the natural infection antibody response.

### Antigenic clustering analyses
To explore the relationship between antibodies representing lineage-specific natural infection and the antigenicity of lineage-specific PsV, we conducted hierarchical clustering (https://www.hiv.lanl.gov/content/sequence/HEATMAP/heatmap.html) on the natural log-transformed neutralization titer data. PsV neutralization titers were reordered according to serological and antigen dendrograms constructed from the resulting Euclidean distance matrices, with clusters supported by resampling the data to generate 500 pseudo-replicates. The antigen clusters were supported by data from many sera, whereas the serum clusters were supported by data from relatively few antigens; thus, more weight should be given to the definition of antigen clusters which is the primary purpose of this evaluation.

The spatial relationship between the lineage-specific PsV antigens was evaluated using the antigenic cartography algorithm (https://acmacs-web.antigenic-cartography.org/)[62]. We first conducted a series of dummy evaluations to assess dimensionality and map optimization. The antigenic maps contained herein were generated in two dimensions (2D) and using 100 optimizations. For data handling purposes, positive neutralization titers were divided by 5,

and negative titers (<50) were assigned a value of <10. In each antigenic map, the gray grid squares represent 1 antigenic unit (AU), which is equivalent to a 2-fold distance; thus, three grid squares are equivalent to $2^3$ or 8-fold distance. To test the robustness of the resulting antigenic distance estimates, several resampling approaches were taken. Data handling was facilitated by Microsoft Excel and supported by bespoke Visual Basic for Applications code. One approach made use of random selection (without replacement) of the type-specific dataset to create 10 pseudo datasets (iterations) containing 90% of the original data which were then used to create 10 pseudo-replicate antigenic maps to derive mean (95%CI) antigenic distances. Two other approaches made use of oversampling to generate pseudo datasets containing increased numbers of pseudo samples representing the rarer lineages. This was done by sampling (with replacement) proportionately to the structure of the original dataset or disproportionately by increasing the lineage representation to a fixed 50 samples per lineage.

Additional corroborative hierarchical clustering matrices and principal component analyses were also performed using a published methodology (https://biit.cs.ut.ee/clustvis/)[63].

### Crystal structure mapping
The pentamer crystal structures of HPV33 (Protein Data Bank [PDB; https://www.wwpdb.org/ and https://www.rcsb.org/] accession number: 6IGE.2), HPV52 (6IGF.1) and HPV58 (5Y9E.1) were used as templates to highlight the location of polymorphic residues between the reference (lineage A) and another lineage within the indicated genotype using Swiss-PdbViewer v4.1 (https://spdbv.unil.ch/).

### Statistical analysis
For lineage antigens to be considered antigenically distinct we set an a priori 4-fold[28,30] distance threshold between the comparison antigen and the reference antigen (lineage A) using antigenic cartography. A sample size estimate for the number of samples required to achieve a power of 80% (1–ß) and a 5% level of significance (α) for detecting a minimum mean difference in neutralization titers of 4-fold between antigens, using the mean (Δ) and standard deviation (σ) of the differences between a variant lineage and the reference lineage for extant data was between 15 and 30 seropositive samples[31,36,64]. We also estimated the number of sample pairs required to detect a significant difference (80% power, 5% significance, 2-sided) between the comparison antigen and the reference antigen with the same difference in magnitude of neutralizing antibody titers as reported previously for HPV52 lineages A/D and HPV58 lineages A/C[31,36] using effect size (Cohen's d; https://statulator.com/SampleSize/ss2PM.html) which resulted in a similar sample size estimate (13–25 samples). Studies of natural infection cohorts have demonstrated that genotype-specific seropositivity is variable but typically around 10–20% for any given genotype[20,22] and so a conservative target of 150 serum samples was sought to represent each lineage-specific infection.

A bespoke study-specific Access® database was created for data management linking uploaded sample information (unique identifier, infecting genotype, sample archive location) to both PsV neutralization titers and metadata once laboratory analyses were completed. All data handling steps were recorded and the data integrity within the database was subjected to vertical audits each time a dataset was uploaded with the outcomes logged.

Differences in seropositivity were assessed using Pearson's Chi-squared ($\chi^2$). We conducted a Shapiro–Wilk test to assess the normality of the distribution of the natural log-transformed titers and a test to assess skewness and kurtosis as well as generating type-specific Q–Q and kernel density plots. We conducted two assessments for each genotype: (i) using all samples regardless of infecting lineage that were positive against lineage A and (ii) only those samples from lineage A

infections that were positive against lineage A. For the former evaluation, all genotypes displayed a non-normal distribution of the natural log-transformed titer data using all statistical tests, while for the latter evaluation, some tests supported a normal distribution for some genotypes. Differences in neutralization titers were therefore assessed using the nonparametric (Wilcoxon sign rank) tests. Tests were 2-tailed and conducted using Stata 17.0 SE (StataCorp., College Station, TX 77845, USA).

### Reporting summary
Further information on research design is available in the Nature Portfolio Reporting Summary linked to this article.

## Data availability
The data that support the findings of this study are available upon request from the IARC (Gary M. Clifford) and the NCI (Aimée R. Kreimer). The data are not publicly available as restrictions apply to the availability of these data, which were used under license for the current study. Data are, however, available upon reasonable request made to the indicated author. De-identified participant data from the Costa Rica Vaccine Trial (CVT) can be shared with outside collaborators for research to understand more about the performance of the HPV vaccine, immune response to the vaccine, and broader study factors associated with the natural history of HPV infection and risk factors for infection and disease. Outside collaborators can apply to access the protocols and data online. For the trial summary, current publications, and contact information for data access see: Human Papillomavirus (HPV) Vaccine Trial in Costa Rica (CVT)–National Cancer Institute (https://dceg.cancer.gov/research/who-we-study/cohorts/costa-rica-vaccine-trial). For the Guanacaste Natural History Study (NHS) summary and contact information for data access see https://dceg.cancer.gov/research/cancer-types/cervix/guanacaste-hpv-natural-history.

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

## Acknowledgements

We are indebted to John T. Schiller and Christopher Buck (National Cancer Institute, Bethesda, MD) for access to the psheLL backbone used for the pseudovirus clones. We thank Busayo Elegunde, Kazutomo Yokoya, and Kavita Panwar (Public Health Microbiology Division, UKHSA) for their technical support and Ross Harris (Data, Analytics Surveillance Department, UKHSA) for helpful discussion.We thank members of the IARC HPV Variant Study Group including previous/current IARC staff (N. Munoz, X. Bosch, S. Franceschi, M. Tommasino, T. Gheit, A. Chen) and local study coordinators in the following countries: Algeria (D. Hammouda), Argentina (D. Loria, E. Matos), Bolivia (J.L. Rios-Dalenz), Brazil (J. Eluf-Neto), Chile (C. Ferreccio, A. Luzoro, J. M. Ojeda, R. Prado), Colombia (N. Aristizabal, L.A. Tafur, M. Molano, H. Posso), Cuba (M. Torroella), Guinea (N. Keita, M. Koulibaly), India (T. Rajkumar, R. Rajkumar), Indonesia (Sarjadi), South Korea (D.-H. Lee, H. R. Shin), Mali (S. Bayo), Morocco (N. Chaouki), Nigeria (J.O. Thomas, C. Okolo, I. Adewole), Panama (E. de los Rios), Paraguay (P.A. Rolon), Peru (E. Caceres, C. Santos), the Philippines (C. Ngelangel), Poland (W. Zatonski), Kenya (P. Gichangi, H. de Vuyst), Spain (S. de Sanjosé, X. Castellsagué), Tanzania (J.N. Kitinya), Thailand (S. Chichareon, S. Sukvirach, S. Tunsakul), Vietnam (TH Anh Tham) and Uganda (H.R. Wabinga). CVT is a long-standing collaboration between investigators in Costa Rica (ACIB-FUNIN) and the NCI. The trial is sponsored and funded by the NCI (contract N01-CP-11005), with funding support from the National Institutes of Health Office of Research on Women's Health. GlaxoSmithKline Biologicals (GSK) provided vaccine and support for aspects of the trial associated with regulatory submission needs of the company under a Clinical Trials Agreement (FDA BB-IND 7920) during the 4-year, randomized blinded phase of our study. We extend a special thanks to the women of Guanacaste and Puntarenas, Costa Rica, who gave of themselves in participating in this effort. This work was supported by the UK Health Security Agency. This work did not receive a specific grant from any funding agency in the public, commercial, or not-for-profit sectors.

## Author contributions

A.G., A.R.K., G.M.C., and S.B. conceived and designed the study. G.M.C., V.T., L.M., M.S., A.R.K., R.H., C.P., and J.T.S. conducted the clinical studies and/or helped with sample identification and retrieval. G.K., F.C.d.S., and A.G. conducted the laboratory testing. G.K., F.C.d.S., V.T., L.M., J.S., and S.B. conducted the data collation and analysis. G.K. and S.B. drafted the manuscript. All authors contributed to the final version of the manuscript.

## Competing interests

The authors declare no competing interests.
