## [Peer Review File · Nature Communications]

Global Evaluation of Lineage-Specific Human Papillomavirus Capsid Antigenicity using Antibodies Elicited by Natural InfectionReviewers' Comments:

Reviewer #1:

Remarks to the Author:

This study explores the antigenic variation among genetic lineages within each of the seven oncogenic genotypes of human papillomavirus. Serum samples from 2,255 women from Africa, the Americas, Asia and Europe who have had a natural infection with one of the seven strains are tested against various lineages of the infecting genotype for all seven oncogenic genotypes. Antigenic profiles for the different lineages were then derived from the antibody data using hierarchical clustering and antigenic cartography. In addition to establishing the antigenic dispersal of lineages within each of seven genotypes this work reports results on experiments with chimeric viruses in which the L1 and L2 gene inserts are swapped between the reference and outlier lineages showing that neutralizing antibodies bind almost exclusively to the major capsid protein.

Overall this is an important piece of work that goes far in quantifying the antigenic variation in HPV. The results are conveyed clearly, and the analysis is robust.

A general comment on this study is that it doesn't explicitly describe certain assumptions or information that lead the authors to test antigenic variation at a particular level in the phylogeny (within accepted genotypes) without testing the antigenic relationship or difference among the accepted genotypes. Just as some forks in the genotype-specific phylogeny are found to be antigenically significant and some not, a priori we may expect some genotypes to be antigenically distant and some not. The fact that the genotypes are considered genotypes and not serotypes suggests that this is not known, and it's not clear for the reader why this piece of work does not test that in the first instance. Perhaps this has been attempted and no cross-reactivity was detected to be able to relate the genotypes to one another. If such attempts have been made the authors should describe or reference them, if the authors have data it would be good to include, if not it would be a very important follow-up study to take a subset of sera (high titre sera) and test cross-genotype reactivity.

If testing sera against strains from other genotypes could yield an antigenic map relating the genotypes to one another it might allow for a more efficient choice of vaccine strains, it might show that some of the outliers described in the study approach the vaccine strain in another genotype as they deviate from their own representative strain, and it would lead to an overall more reliable placement of strains on the map through better triangulation, and it would improve the effect of scarcity of sera in certain subtypes.

It is good that the authors test the potential confounding of co-infections on serum reactivity. They could mention that two potential explanations for the robustness of the results to removal of these sera are possible: it could be that the multi-infection sera are heavily outnumbered, and it could be that the co-infecting strains don't interact with one another.

The outlier HPV58 C could be a low reactive virus and not as antigenically distant as it seems given it also has low reaction with its homologous serum. In this case it would be worth adding a caveat that HPV58 C is reading low to all sera and to mention the possibility of a low reactivity issue for this particular pseudotype. The authors might also comment here that there are two non-HPV58 C sera that are close to the C antigen in the map, and why this might be.

In the case of HPV45 for which only two variants are available the authors have not produced antigenic maps. Such maps would be possible, yes, each serum position could not be resolved from being on one side of the two antigens or the other, but since the authors are primarily considering antigen distance they should still be able to resolve the antigenic distance.

The imbalance in available sera within a genotype could be ameliorated through bootstrapping to get approximately the same number of sera for each of the lineages within a genotype, this would be a good robustness test.

Line 101: "Africa (575; 25%), The Americas (1270; 56%), Asia (373; 17%), Europe (37; 2%) and Oceania (0; 102 0%)."

The order of Africa and The Americas could be reversed to yield an overall descending order.

Reviewer #2:

Remarks to the Author:

Authors have presented a manuscript ' Global Evaluation of Lineage-Specific Human Papillomavirus Capsid Antigenicity using Antibodies Elicited by Natural Infection ' to be considered for publication in the journal Nature Communications.

The article has a novel idea to assess antigenic variation for each oncogenic HPV type 16, 18, 31, 33, 45, 52 and 58 at variant level, which would likely reveal if there is variant-dependent differential antibody specific immune responses. Although some antigenic variation is observed in this study there are crucial limitations to the work.

1. A major issue is that the genetic ancestry of the serum/plasma sample used for the analysis of differential variant level antigenic variation is not explicitly tested, although the seropositivity rate has a significant geographic component eg. for HPV16. Hence, the authors should soften their conclusions particularly regarding geographical interpretations unless the ancestry of the individuals behind the samples is measured using molecular ancestry markers.

2. More than half of the serum (or plasma) samples (56%) were from the Americas and only 2% from Europe. This means that there is a strong geographical sampling bias (surrogate) in this dataset, while HPVs are known to have an extremely large and diverse geographical standing population diversity of the viral lineages. Hence, authors should sensitivity test and describe what part this sampling bias may play in the results, if any.

1. Indeed, it would be helpful for the authors to estimate what role/outcome the differential seropositivity rate might have played for the type-level variant results eg. for the HPV16 type variants.

2. Fig. 2 shows no fold differences in neutralization titers for HPV16 lineage specific sera. However, the authors write, "These differential neutralizing antibody titers manifest as reduced seropositivity against the reference lineage A antigen in some cases; for example, sera positive for neutralizing antibodies against outlier lineage antigens HPV16 C (50/64; 78%), HPV33 B (5/9; 56%), HPV52 D (6/9; 67%) and HPV58 C (5/6; 83%) demonstrated reduced seropositivity against lineage A antigens, as indicated." Authors should address the likely discrepancy with the above sentences in the main text.

Minor comments:

- Figure 2. regarding the hierarchical clustering of neutralizing antibody titers the plot is not that readable, please explore other summary statistics and ordination methods to improve the visualization and testing of the interactions.

- Lineage-specific antigenic cartographs need a better visual resolution and a more detailed figure description, currently they are not that reader friendly.

We thank the referees for their time and consideration in reviewing our manuscript “Global Evaluation of Lineage-Specific Human Papillomavirus Capsid Antigenicity using Antibodies Elicited by Natural Infection” and believe that the revisions suggested by the reviewers have greatly improved the manuscript. Our responses to the comments raised are highlighted in blue below.

Reviewer #1:

This study explores the antigenic variation among genetic lineages within each of the seven oncogenic genotypes of human papillomavirus. Serum samples from 2,255 women from Africa, the Americas, Asia and Europe who have had a natural infection with one of the seven strains are tested against various lineages of the infecting genotype for all seven oncogenic genotypes. Antigenic profiles for the different lineages were then derived from the antibody data using hierarchical clustering and antigenic cartography. In addition to establishing the antigenic dispersal of lineages within each of seven genotypes this work reports results on experiments with chimeric viruses in which the L1 and L2 gene inserts are swapped between the reference and outlier lineages showing that neutralizing antibodies bind almost exclusively to the major capsid protein. Overall, this is an important piece of work that goes far in quantifying the antigenic variation in HPV. The results are conveyed clearly, and the analysis is robust.

We thank the reviewer for his/her considered review of the manuscript and positive comments.

A general comment on this study is that it doesn't explicitly describe certain assumptions or information that lead the authors to test antigenic variation at a particular level in the phylogeny (within accepted genotypes) without testing the antigenic relationship or difference among the accepted genotypes. Just as some forks in the genotype-specific phylogeny are found to be antigenically significant and some not, a priori we may expect some genotypes to be antigenically distant and some not. The fact that the genotypes are considered genotypes and not serotypes suggests that this is not known, and it's not clear for the reader why this piece of work does not test that in the first instance. Perhaps this has been attempted and no cross-reactivity was detected to be able to relate the genotypes to one another. If such attempts have been made the authors should describe or reference them, if the authors have data it would be good to include, if not it would be a very important follow-up study to take a subset of sera (high titre sera) and test cross-genotype reactivity. If testing sera against strains from other genotypes could yield an antigenic map relating the genotypes to one another it might allow for a more efficient choice of vaccine strains, it might show that some of the outliers described in the study approach the vaccine strain in another genotype as they deviate from their own representative strain, and it would lead to an overall more reliable placement of strains on the map through better triangulation, and it would improve the effect of scarcity of sera in certain subtypes.

We agree that the Introduction section could have benefitted from inclusion of some additional historical context on inter-genotype diversity, particularly in relation to antigenicity. Analysis of HPV genomic sequences have led to estimates for the phylogenetic emergence of specific types from their common ancestors around 5 – 20 million years ago, with further evolutionary branches being resolved at around 200 – 500 thousand years ago for lineages and 50 – 200 thousand years ago for sublineages [1, 2]. These levels within the papillomavirus phylogeny have traditionally been resolved by sequencing the L1 ORF (more recently by whole genome sequencing); thus, the term ‘type’ is used to describe one level of genetic diversity within the phylogeny [3].

The HPV field does not traditionally describe types in terms of serological responses against its antigens (i.e., classification by serotype), unlike for some other viruses. However, important early capsid immunogenicity work in mice and rabbits demonstrated a predominant type-specific neutralizing antibody response, with some minor cross-reactivity between related types [4, 5]. We have previously published a pre-clinical immunogenicity study

using capsids representing a broad range of types belonging to the alpha-9 and alpha-7 species groups and demonstrated a similar majority type-specific response with some low level cross-reactivity between related types in keeping with their genetic distance profile [6]. In response to this reviewer's comments, we have used these published data to create relational antigenic maps (see Figure R1, below) for a direct comparison with the data included in this manuscript. This analysis shows the tight clustering of individual rabbit sera around their respective antigens and the orders of magnitude greater distance between antigens at the level of type compared to the lineage level described in this manuscript. For example, the closest inter-type relationship found was a 100-fold distance between HPV33 and HPV58 antigens, compared to the 2-20-fold inter-lineage distances found using lineage-specific natural infection sera evaluated in this manuscript (cf., Figure 4). Thus, at least at the level of type within the phylogeny, genotypes are equivalent to serotypes.

Figure R1. Antigenic maps based upon rabbit type-specific antisera [6]. Left panel, Alpha-9 types (HPV16, 18, 31, 33, 35, 52 and 58) and right panel, Alpha-7 types (HPV18, 39, 45, 59 and 68). Filled circles and open squares represent type-specific pseudovirus antigens and rabbit sera (n=3 per type), respectively. In each antigenic map the grey grid squares represent 1 antigenic unit (AU), which is equivalent to a 2-fold difference; thus, three grid squares is equivalent to an 8-fold (2^3) difference.

It is also apparent that the HPV prophylactic L1 capsid based vaccines elicit some degree of functional cross-reactive antibody response in humans as we and others have highlighted [7-9] and it is reasonably assumed that this cross-reactivity measured in vitro is at least an indicator of the cross-protective capacity of the HPV vaccines [10]. It should be noted, however, that the cross-reactive antibody response is at least two orders of magnitude lower than that generated against the type-specific antigen and may not be as durable as type-specific immunity [8, 9, 11]. Cross-protection is therefore unlikely to be as robust as direct protection [12].

These cross-reactive antibody specificities are apparent when sera containing the high levels of antibodies elicited following the immunization of small animals or following the vaccination of humans are evaluated. It is difficult to imagine being able to measure such cross-reactive antibodies in the serum of individuals following natural infection where the antibody levels are orders of magnitude lower and there would always remain some uncertainty about the infection history of the individuals from which the samples would be derived.

We have added some text to the manuscript (lines 61-64, 76-78, 183-186 and 256-258) and have included the previously unpublished antigenic maps derived using the pre-clinical sera data in the Supplementary File. We believe that these additions address the concerns of this reviewer appropriately and improve the description of the context in which this current study is set.

It is good that the authors test the potential confounding of co-infections on serum reactivity. They could mention that two potential explanations for the robustness of the results to removal of these sera are possible: it could be that the multi-infection sera are heavily outnumbered, and it could be that the co-infecting strains don't interact with one another.

We agree with the reviewer that there are different explanations for the robustness of the results following removal of data representing individuals with evidence of mixed infection, specifically: (i) that there is a biological mechanism for the antibodies elicited following natural infection with multiple HPV types to interfere with the estimation of antigenic distance between specific types but in this case the low number of mixed infection samples was insufficient to bestow interference of a measurable magnitude or (ii) that there is no such biological mechanism and the type-specific antibody responses are independent and do not interfere with one another. We have added text lines 252-256 to the manuscript on this point.

The outlier HPV58 C could be a low reactive virus and not as antigenically distant as it seems given it also has low reaction with its homologous serum. In this case it would be worth adding a caveat that HPV58 C is reading low to all sera and to mention the possibility of a low reactivity issue for this particular pseudotype.

We are mindful of the possibility of intrinsic and/or technical issues with individual antigens that may inadvertently affect the interpretation of antibody specificity and so we go to great lengths to address this by standardizing pseudovirus infectivity input into the neutralization assay, evaluating particle morphology (for example [13]), generating within-study quality assurance data and where appropriate creating chimeric PsV to address the context of indicated capsid sequences. There is no evidence that the HPV58 C should be considered technically different from any other PsV that we have used in this or other studies, but rather represents a lower specific susceptibility to lineage A antibodies due to lineage-specific amino acid changes in the L1 capsid protein as we have previously shown using monoclonal antibodies, nonavalent vaccine and natural infection sera [13-15].

Neutralizing antibody titers following natural infection are low in magnitude compared to those elicited following pre-clinical immunization or vaccination. In this case, antibodies elicited following HPV58 A infection reacted with its homologous lineage A antigen similarly to the reactivity of antibodies elicited following HPV58 lineage C infection against its homologous lineage C antigen ($p=0.102$; Mann Whitney U test; Supplementary Table 2). We have previously evaluated the antigenicity of HPV58 capsids using lineage-specific murine antisera finding that both A and C antigens generated titers of $\sim 100,000$ against their homologous antigen which was reduced by $\sim 2 \text{ Log}_{10}$ when tested against the heterologous antigen [13]. The data support reciprocal differences in antibody susceptibility between lineages A and C. We have also presented data in the current manuscript using natural infection sera and in previous publications using sera and monoclonal antibodies [13, 14] demonstrating that chimeric PsV expressing the L1 or fragments thereof (e.g., DE, FG and/or HI loops) of the HPV58 lineages A or C L1 capsids appropriately reciprocate HPV58 lineage A and lineage C antibody specificities.

These data support HPV58 A and C lineage pseudoviruses being antigenically distinct due to specific amino acid residues on the L1 capsid surface, rather than a low reactivity issue in general. We have already cited examples of our evaluations of lineage-specific reactivity in the text (263-266) but have added a line specifically about the reciprocity of the murine lineage-specific antisera (lines 266-267).

The authors might also comment here that there are two non-HPV58 C sera that are close to the C antigen in the map, and why this might be.

The reviewer points out an important feature of these antigenic maps. In some cases, not limited to the HPV58 lineage C example highlighted by the reviewer, an individual serum can be placed nearer to the coordinates of a heterologous antigen than its lineage-specific homologous counterpart. These observations are due to differences in the individual level antibody specificities that, in some cases, a serum can react to a heterologous antigen with a similar or higher antibody titer than against its homologous antigen. The position of the antigens is estimated following a number of iterations which take into account the reactivity of all the sera against all the antigens. We have added some text on this in the figure legend to Figure 4.

In the case of HPV45 for which only two variants are available the authors have not produced antigenic maps. Such maps would be possible, yes, each serum position could not be resolved from being on one side of the two antigens or the other, but since the authors are primarily considering antigen distance they should still be able to resolve the antigenic distance.

To address the inability to accurately map HPV45 lineage A and B antigens, we simulated separate antigenic maps which included an additional dataset using an exact copy of lineage A (labelled A') or lineage B (labelled B') data (Figure R2). This approach permitted an estimate of antigenic distance between lineage A and B antigens of 1.8-fold for both evaluations. A statement on this outcome has been added to the text in lines 171-174 and the maps below added to the Supplementary File.

Figure R2. Antigenic maps of HPV45 lineages A and B. To estimate the antigenic distance between antigens A and B, a dummy dataset was included in each map using a copy of lineage A (A', left panel) or lineage B data (B', right panel).

The imbalance in available sera within a genotype could be ameliorated through bootstrapping to get approximately the same number of sera for each of the lineages within a genotype, this would be a good robustness test.

This imbalance is particularly evident for some of the rarer lineages of types HPV33, HPV52 and HPV58 where the expected 15 – 30 seropositive samples per lineage was not reached (Supplementary Table 2). There is therefore a valid concern that the reported antigenic distances may not be representative of the population as the samples tested may represent a skewed subset due to the low representation of the rarer lineages.

The term “bootstrapping” is commonly used to describe a process whereby multiple pseudo-datasets of usually the same or smaller size are created by resampling data (usually with replacement) from the original dataset in order to derive summary statistics of the sampled population. However, this approach does not increase the size of the dataset per se, only the amount of data available to use in estimates of the original sample distribution. We had already conducted a randomized resampling evaluation (without replacement) to test the robustness of the antigenic distance estimations by creating 10 pseudo-datasets with a sample size of 90% of the original dataset to estimate the mean (95% CI) antigenic distance (Supplementary Figure 3).

To address the reviewer’s comments, we have conducted randomized resampling (with replacement) evaluations with oversampling in order to create pseudo-datasets larger than the original sample size resulting in an increased number of samples representing each lineage (Figure R3). The disadvantage of this approach is that for lineages where the number of original samples is low, samples are more likely to be used repeatedly which is more likely to distort findings on the basis of chance (as with any small sample) and result in “lumpy” distributions. An alternative approach would be to derive artificial pseudo-samples on the basis of some modelled approach, using a model of the distribution and relationships between the serological measures. Parametric assumptions – such as least squares regression and related methods – may over-simplify the data structure and lead to biased estimates of mean distances. Classification and regression tree (CART) analysis was explored to generate new pseudo-samples which offered a more faithful reproduction of the observed sera distributions but did little to ameliorate the “lumpy” distributions where data were sparse. We therefore opted for the simpler bootstrap approach, which although not ideal, does use real data to provide further sensitivity analyses that support the robustness of our conclusions.

We undertook two such evaluations. First, a matrix representing each type was created and populated by sample data randomly selected (with replacement) from the original type-specific dataset. The matrix size was n=150 samples for HPV33 and n=200 samples for HPV52 and HPV58. As expected, this approach boosted the numbers of samples overall by about 2-fold and as this was done proportionately across each dataset the structure of the original dataset should be retained in the final matrix. The effect of this approach was to increase the apparent number of samples representing some of the rarer lineages so that they met or were closer to the a priori sample size estimate of 15 – 30 seropositive samples per lineage. This was repeated n=10 times for each type to generate mean (95% CI) antigenic distances. Next, a matrix representing each type was created and populated by a fixed amount of sample data (n=50 per lineage) randomly selected (with replacement) from the lineage-specific data within the original dataset. The matrix size was also n=150 samples for HPV33 and n=200 samples for HPV52 and HPV58. This approach would increase the number of pseudo samples disproportionately and therefore there was no expectation that the resulting matrix will retain the structure of the original dataset. Thus, lineage A would be represented by fewer samples than in the original dataset and other lineages will have samples represented multiple times and will exhibit a more uneven distribution. This was

repeated $n=10$ times for each type to generate mean (95% CI) antigenic distances. These approaches highlighted the same outlier lineages as in the original assessment suggesting that the study outcomes are robust. The proportionate resampling evaluation which attempted to retain the structure of the original dataset generated antigenic distance estimates very close to the original values, while the second evaluation highlighted the same lineages as being outliers, but the distance estimates were slightly different likely due to the disproportionate representation of replicated values for each lineage (Figure 4).

Overall, these data provide further support for the distinct antigenicity of the outlier lineages, despite the overall shortfall in sample numbers for some of the rarer lineages. We have included a description of these evaluations in the manuscript text (lines 200-207 and 315-318) and added the figure to the Supplementary File.

Method	Type	N	Fold	B	C	D
Original dataset	HPV33	A=69 B=10 C=4	A	4.2	5.9	
			B		1.7	
			C			
	HPV52	A=71 B=17 C=18 D=10	A	2.3	1.1	3.9
			B		2.2	3.5
			C			3.6
	HPV58	A=63 B=9 C=8 D=4	A	1.7	17.2	1.2
			B		15.7	1.7
			C			19.8
Proportionate randomized resampling with replacement	HPV33	A=127 B=17 C=7	A	4.1 (3.8 – 4.4)	5.9 (5.5 – 6.3)	
			B		1.8 (1.7 – 1.8)	
			C			
	HPV52	A=125 B=28 C=29 D=18	A	2.2 (2.2 – 2.3)	1.1 (1.0 – 1.1)	3.8 (3.6 – 3.9)
			B		2.2 (2.1 – 2.3)	3.6 (3.4 – 3.8)
			C			3.6 (3.5 – 3.7)
	HPV58	A=147 B=23 C=20 D=10	A	1.7 (1.6 – 1.8)	16.2 (14.7 – 17.7)	1.2 (1.2 – 1.3)
			B		14.5 (12.8 – 16.2)	1.7 (1.6 – 1.8)
			C			18.9 (17.0 – 20.9)
Disproportionate randomized resampling with replacement	HPV33	A=50 B=50 C=50	A	4.3 (4.1 – 4.5)	4.8 (4.6 – 5.0)	
			B		1.5 (1.4 – 1.5)	
			C			
	HPV52	A=50 B=50 C=50 D=50	A	2.3 (2.3 – 2.4)	1.1 (1.0 – 1.1)	4.1 (3.9 – 4.3)
			B		2.3 (2.2 – 2.4)	4.0 (3.8 – 4.3)
			C			3.9 (3.8 – 4.1)
	HPV58	A=50 B=50 C=50 D=50	A	1.8 (1.8 – 1.9)	8.1 (7.6 – 8.6)	1.3 (1.3 – 1.4)
			B		6.2 (5.9 – 6.5)	1.8 (1.7 – 1.9)
			C			9.5 (8.9 – 10.2)

Figure R3. Antigenic distance estimates following random resampling with replacement evaluations. Estimates of antigenic distance (mean, 95%CI) between lineage antigens following randomized resampling with replacement. Resampling was proportionate by random selection from the type-specific dataset where N represents the mean number of samples for each lineage after 10 iterations or disproportionate where each lineage was resampled until a target of N=50 samples for each lineage was reached.

Line 101: “Africa (575; 25%), The Americas (1270; 56%), Asia (373; 17%), Europe (37; 2%) and Oceania (0; 102 0%).” The order of Africa and The Americas could be reversed to yield an overall descending order.

We have amended the statement in the text (line 105) to reflect the descending order of the number samples available from these regions.

Reviewer #2:

Authors have presented a manuscript 'Global Evaluation of Lineage-Specific Human Papillomavirus Capsid Antigenicity using Antibodies Elicited by Natural Infection' to be considered for publication in the journal Nature Communications. The article has a novel idea to assess antigenic variation for each oncogenic HPV type 16, 18, 31, 33, 45, 52 and 58 at variant level, which would likely reveal if there is variant-dependent differential antibody specific immune responses. Although some antigenic variation is observed in this study there are crucial limitations to the work.

We thank the reviewer for his/her considered review of the manuscript and positive comments.

1. A major issue is that the genetic ancestry of the serum/plasma sample used for the analysis of differential variant level antigenic variation is not explicitly tested, although the seropositivity rate has a significant geographic component e.g. for HPV16. Hence, the authors should soften their conclusions particularly regarding geographical interpretations unless the ancestry of the individuals behind the samples is measured using molecular ancestry markers.

We agree with the reviewer that the geographical source of these samples should not imply a specific genetic ancestry or ethnicity of these individuals. Although we have cited sources linking the dispersal of lineages to geography, we have made no such assumption about the relationship between geography and ethnicity. In fact, we were careful not to make such assertions by referring to the samples as being collected from 'women resident in Africa, The Americas...' etc., we were simply stating the geographic source of the samples. This caution is warranted because it is clear that individual lineages are differentially distributed globally and can be overrepresented in high grade disease and while the role of ethnicity in differential disease outcomes is unclear, at least for some sublineages of HPV16 there does seem to be an association [16]. In this study, serostatus was associated with geographical region ($p=0.001$) for only one genotype studied, HPV16. Even after removal of the underrepresented samples from Europe this association remained ($p=0.023$) seemingly due to a lower seropositivity rate from the Asian region.

In response to this reviewer, we have explicitly stated that by referring to samples being collected from women resident in these regions we are not inferring their genetic ancestry or ethnicity (lines 385-386).

2. More than half of the serum (or plasma) samples (56%) were from the Americas and only 2% from Europe. This means that there is a strong geographical sampling bias (surrogate) in this dataset, while HPVs are known to have an extremely large and diverse geographical standing population diversity of the viral lineages. Hence, authors should sensitivity test and describe what part this sampling bias may play in the results, if any.

We agree with the reviewer that despite the large-scale collection of samples for this study there was unfortunately a significant geographical sampling bias. We specifically highlighted this issue as a shortcoming in the manuscript, but we are happy to address it further. In response to this specific comment, we conducted a sensitivity analysis wherein we removed data associated with the samples from Europe and one other geographical region (Africa, The Americas or Asia) and estimated the antigenic distances using antigenic cartography. We conducted this assessment using data for HPV16 as this type had the most samples with all lineages represented by the three major regions and was therefore amenable to such an assessment (Supplementary Figure 1) but also because this type has been quite rightly highlighted by the reviewer as exhibiting some geographical bias. We conducted three such assessments removing a different major geographical region (Africa, The Americas or Asia) each time and compared the outcome distances to those generated using the whole dataset (Figure R4). Overall, the distances between lineage-specific antigens were similar and the distance between lineage A and C antigens was maintained at between 2 – 4-fold, though it is possible that removal of samples from Africa (enriched for lineages B and C) may have had some influence.

Sample set	Fold	B	C	D
Whole dataset (n=226 seropositive samples)	A	1.2	2.7	1.9
	B		2.6	1.7
	C			2.0
Asia and the Americas (n=87; no African or European samples)	A	1.3	2.3	1.9
	B		2.0	1.4
	C			1.6
Africa and Asia (n=164; no samples from the Americas or Europe)	A	1.1	2.8	1.9
	B		2.8	1.8
	C			2.2
African and the Americas (n=189; no samples from Asia or Europe)	A	1.3	2.9	2.1
	B		2.7	1.7
	C			2.1

Figure R4. Sensitivity analysis of HPV16 antigenic distance estimates. Estimates of antigenic distance between lineage antigens following removal of samples from indicated geographical regions.

While the number of samples from Europe in this study is very small, we have previously examined lineage-specific pseudoviruses of HPV52 and HPV58 using sera collected from women residing in Europe (specifically, Italy) and for whom, where measured, the majority represented lineage A infections [13, 17]. In these studies, HPV52 lineage D and HPV58 lineage C pseudoviruses were less sensitive compared to their equivalent lineage A pseudoviruses. We have added the outcome from the sensitivity analysis to the Supplementary file and updated the manuscript text to include a description of this analysis (lines 194-200 and 312-318).

1. Indeed, it would be helpful for the authors to estimate what role/outcome the differential seropositivity rate might have played for the type-level variant results e.g. for the HPV16 type variants.

Higher rates of seropositivity would generate a larger sample dataset with which to use for antigenic mapping and therefore would allow better precision in the estimates of antigenic distance. In the manuscript we attempted to highlight the issue of precision by conducting a small series of iterations using a 90% resampling method for each type to generate mean (95%CI) antigenic distance estimates (Supplementary File). The estimate of precision for HPV16 gave a mean distance of 2.7-fold (95%CI, 2.6 – 2.7) between lineage antigens A and C.

In response to the comment by this reviewer highlighting the differential serostatus rate for HPV16 (although this was manifest for the variable [Region] and not the variable [Lineage]), we estimated antigenic distances using a randomized 50% smaller seropositive sample for each lineage compared to the full dataset to simulate a significantly lower seropositivity rate. Removal of 50% (range 41 – 62%) of seropositive samples had little or no impact on the estimated antigenic distances demonstrating that these estimates are robust (Figure R5).

Sample set	Fold	B	C	D
Whole dataset (n=226 seropositive samples)	A	1.2	2.7	1.9
	B		2.6	1.7
	C			2.0
Reduced lineage A samples (n=27/46; n=207 total samples)	A	1.2	2.6	1.9
	B		2.6	1.6
	C			2.0
Reduced lineage B samples (n=19/50; n=195 total samples)	A	1.2	2.6	2.0
	B		2.5	1.6
	C			1.9
Reduced lineage C samples (n=32/64; n=194 total samples)	A	1.3	2.5	1.9
	B		2.4	1.6
	C			1.8
Reduced lineage D samples (n=30/66; n=190 total samples)	A	1.2	2.7	2.0
	B		2.6	1.7
	C			2.1

Figure R5. Impact estimate of lower rates of seropositivity. Estimates of antigenic distance between lineage antigens following removal of ca. 50% of seropositive samples representing indicated lineages.

We have added the outcome from this analysis to the Supplementary file and updated the manuscript text to include a description of this analysis (lines 194-200 and 312-318).

2. Fig. 2 shows no fold differences in neutralization titers for HPV16 lineage specific sera. However, the authors write, “These differential neutralizing antibody titers manifest as reduced seropositivity against the reference lineage A antigen in some cases; for example, sera positive for neutralizing antibodies against outlier lineage antigens HPV16 C (50/64; 78%), HPV33 B (5/9; 56%), HPV52 D (6/9; 67%) and HPV58 C (5/6; 83%) demonstrated reduced seropositivity against lineage A antigens, as indicated.” Authors should address the likely discrepancy with the above sentences in the main text.

HPV16 lineage C sera exhibited lower neutralizing antibody titers against lineage A, B and D pseudoviruses compared to the HPV16 C lineage antigen (Supplementary Table 2 and blue-filled box and whisker plot in Figure 2). These lower neutralization titers also manifest as reduced seropositivity rates against these antigens with, for example, only 50 of 64 (78%; 95%CI, 66 – 85%) lineage C positive serum samples being positive against the lineage A pseudovirus (Supplementary Table 2). We have amended the text for clarity and limited the example given to HPV16 where the number of lineage specific positive samples is high enough to be confident in the point estimate of lower seropositivity (lines 135-138).

Minor comments:

- Figure 2. regarding the hierarchical clustering of neutralizing antibody titers the plot is not that readable, please explore other summary statistics and ordination methods to improve the visualization and testing of the interactions.

The purpose of the hierarchical clustering was to evaluate whether these functional data can be used to identify lineage-specific antigen clusters by providing an outcome supported by a high proportion of resampling iterations (bootstraps). This approach has been used by ourselves [18] and others [19-21] for a range of targets including HPV, Influenza virus, HIV and SARS-CoV-2 to establish segregation of antigens and/or immune responses and we believe represents an important tool with which to evaluate of lineage-specific antigenicity. We have addressed the reviewers concerns in the following ways:

We have improved the resolution of the individual heatmaps and that of the final image, have replaced the indicative bootstrap value (***) with the actual percentage (100%) of iterations and have improved the figure legend for the reader to better understand how to interpret these plots.

We have also employed additional approaches to corroborate these data using published methodologies (ClustVis [22]) (Figure R6). Initially, we corroborated the segregation of antigens using heatmaps supported by Euclidean distance dendrograms. These estimates were not underpinned by bootstrapping but nevertheless corroborate the separation of antigens found in Figure 3. It is possible to see some enrichment of lineage-specific serum samples within some serum clusters (e.g., HPV16C, HPV33B, HPV52D and HPV58C) but without bootstrap support these observations can only be anecdotal. In addition, we made use of the principal component analysis algorithm with the ClustVis programme [22]. The PC1 and PC2 channels accounted for the majority of the variance across these datasets as indicated: HPV16 (91%), HPV18 (100%), HPV31 (100%), HPV33 (100%), HPV45 (100%), HPV52 (97%), HPV58 (98%). These data corroborate the separation of antigens by use of hierarchical clustering but also corroborate the relative coordinates of the antigens in two-dimensional space found with the antigenic cartography analysis (Figure 4). Taken together, these algorithms provide strong support for the differential antigenicity of lineage-specific capsid antigens within each genotype examined.

We have added these figures to the Supplementary File and have added appropriate text to the manuscript (lines 150-154, 179-181, 309-310, 439-440).

Figure R6. Use of additional clustering techniques to support lineage-specific antigenicity. Top panel, hierarchical clustering and heatmap using indicated relative scale. Natural log neutralizing antibody titers were reordered according to serological and antigen dendrograms constructed from the resulting Euclidean distance matrices. Serum side bar denotes natural infection lineage from which serum (or plasma) sample derived according to key. Bottom panel, principal component analysis to define relative antigen position in two-dimensional space. The PC1 and PC2 channels accounted for the majority of the variance across these datasets as indicated: HPV16 (91%), HPV18 (100%), HPV31 (100%), HPV33 (100%), HPV45 (100%), HPV52 (97%), HPV58 (98%). Both analyses made use of the ClustVis program. [22].

- Lineage-specific antigenic cartographs need a better visual resolution and a more detailed figure description, currently they are not that reader friendly.

We have improved resolution of the maps within the figure and provided a more detailed figure legend.

References

- [1] Pimenoff VN, de Oliveira CM Bravo IG. Transmission between Archaic and Modern Human Ancestors during the Evolution of the Oncogenic Human Papillomavirus 16. *Mol Biol Evol*. 2017; 34: 4-19.
- [2] Chen Z, Ho WCS, Boon SS, Law PTY, Chan MCW, DeSalle R, Burk RD, Chan PKS. Ancient Evolution and Dispersion of Human Papillomavirus 58 Variants. *J Virol*. 2017; 91: e01285-17.
- [3] de Villiers EM, Fauquet C, Broker TR, Bernard HU, zur Hausen H. Classification of papillomaviruses. *Virology*. 2004; 324: 17-27.
- [4] Giroglou T, Sapp M, Lane C, Fligge C, Christensen ND, Streeck RE, Rose RC. Immunological analyses of human papillomavirus capsids. *Vaccine*. 2001; 19: 1783-93.
- [5] Ochi H, Kondo K, Matsumoto K, Oki A, Yasugi T, Furuta R, Hirai Y, Yoshikawa H, Kanda T. Neutralizing antibodies against human papillomavirus types 16, 18, 31, 52, and 58 in serum samples from women in Japan with low-grade cervical intraepithelial neoplasia. *Clin Vaccine Immunol*. 2008; 15: 1536-40.
- [6] Bissett SL, Mattiuzzo G, Draper E, Godi A, Wilkinson DE, Minor P, Page M, Beddows S. Pre-clinical immunogenicity of human papillomavirus alpha-7 and alpha-9 major capsid proteins. *Vaccine*. 2014; 32: 6548-55.
- [7] Bissett SL, Godi A, Jit M, Beddows S. Seropositivity to non-vaccine incorporated genotypes induced by the bivalent and quadrivalent HPV vaccines: A systematic review and meta-analysis. *Vaccine*. 2017; 35: 3922-9.
- [8] Stanley M, Joura E, Yen GP, Kothari S, Luxembourg A, Saah A, Wailia A, Perez G, Khoury H, Badgley D, Brown DR. Systematic literature review of neutralizing antibody immune responses to non-vaccine targeted high-risk HPV types induced by the bivalent and the quadrivalent vaccines. *Vaccine*. 2021; 39: 2214-23.
- [9] Mariz FC, Gray P, Bender N, Eriksson T, Kann H, Apter D, Paavonen J, Pajunen E, Prager KM, Sehr P, Surcel HM, Waterboer T, Miller M, Pawlita M, Lehtinen M. Sustainability of neutralising antibodies induced by bivalent or quadrivalent HPV vaccines and correlation with efficacy: a combined follow-up analysis of data from two randomised, double-blind, multicentre, phase 3 trials. *Lancet Infect Dis*. 2021; 21: 1458-68.

- [10] Schiller J, Lowy D. Explanations for the high potency of HPV prophylactic vaccines. *Vaccine*. 2018;36:4768-73.
- [11] Draper E, Bissett SL, Howell-Jones R, Waight P, Soldan K, Jit M, Andrews N, Miller E, Beddows S. A randomized, observer-blinded immunogenicity trial of Cervarix((R)) and Gardasil((R)) Human Papillomavirus vaccines in 12-15 year old girls. *PLoS One*. 2013;8:e61825.
- [12] Brown DR, Joura EA, Yen GP, Kothari S, Luxembourg A, Saah A, Walia A, Perez G, Khoury H, Badgley D, Stanley M. Systematic literature review of cross-protective effect of HPV vaccines based on data from randomized clinical trials and real-world evidence. *Vaccine*. 2021;39:2224-36.
- [13] Godi A, Martinelli M, Haque M, Li S, Zhao Q, Xia N, Cocuzza CE, Beddows S. Impact of Naturally Occurring Variation in the Human Papillomavirus (HPV) 58 Capsid Proteins on Recognition by Type-Specific Neutralizing Antibodies. *J Infect Dis*. 2018;218:1611-21.
- [14] Godi A, Boampong D, Elegunde B, Panwar K, Fleury M, Li S, Zhao Q, Xia N, Christensen ND, Beddows S. Comprehensive Assessment of the Antigenic Impact of Human Papillomavirus Lineage Variation on Recognition by Neutralizing Monoclonal Antibodies Raised Against Lineage A Major Capsid Proteins of Vaccine-Related Genotypes. *J Virol*. 2020;94:e01236-20.
- [15] Godi A, Kemp TJ, Pinto LA, Beddows S. Sensitivity of Human Papillomavirus (HPV) Lineage and Sublineage Variant Pseudoviruses to Neutralization by Nonavalent Vaccine Antibodies. *J Infect Dis*. 2019;220:1940-5.
- [16] Mrabello L, Yeager M, Cullen M, Boland JF, Chen Z, Wentzensen N, Zhang X, Yu K, Yang Q, Mitchell J, Roberson D, Bass S, Xiao Y, Burdett L, Raine-Bennett T, Lorey T, Castle PE, Burk RD, Schiffman M. HPV16 Sublineage Associations With Histology-Specific Cancer Risk Using HPV Whole-Genome Sequences in 3200 Women. *J Natl Cancer Inst*. 2016;108.
- [17] Godi A, Bissett SL, Masloh S, Fleury M, Li S, Zhao Q, Xia N, Cocuzza CE, Beddows S. Impact of naturally occurring variation in the human papillomavirus 52 capsid proteins on recognition by type-specific neutralising antibodies. *J Gen Virol*. 2019;100:237-45.
- [18] Bissett SL, Draper E, Myers RE, Godi A, Beddows S. Cross-neutralizing antibodies elicited by the Cervarix(R) human papillomavirus vaccine display a range of Alpha-9 inter-type specificities. *Vaccine*. 2014;32:1139-46.
- [19] Binley JM, Win T, Korber B, Zwick MB, Wang M, Chappey C, Stiegler G, Kunert R, Zolla-Pazner S, Katinger H, Petropoulos CJ, Burton DR. Comprehensive cross-clade neutralization analysis of a panel of anti-human immunodeficiency virus type 1 monoclonal antibodies. *J Virol*. 2004;78:13232-52.
- [20] Dugan HL, Guthmiller JJ, Arevalo P, Huang M, Chen YQ, Neu KE, Henry C, Zheng NY, Lan LY, Tepora ME, Stovicek O, Bitar D, Palm AE, Stamper CT, Changrob S, Uset HA, Coughlan L, Krammer F, Cobey S, Wilson PC. Preexisting immunity shapes distinct antibody landscapes after influenza virus infection and vaccination in humans. *Science translational medicine*. 2020;12.
- [21] Selva KJ, van de Sandt CE, Lenke MM, Lee CY, Shoffner SK, Chua BY, Davis SK, Nguyen THO, Rowntree LC, Hensen L, Koutsakos M, Wong CY, Mordant F, Jackson DC, Flanagan KL, Crowe J, Tosif S, Neeland MR, Sutton P, Licciardi PV, Crawford NW, Cheng AC, Doolan DL, Amanat F, Krammer F, Chappell K, Muthirani N, Watterson D, Young P, Lee WS, Wines BD, Mark Hogarth P, Esterbauer R, Kelly HG, Tan HX, Juno JA, Wheatley AK, Kent SJ, Arnold KB, Kedzierska K, Chung AW. Systems serology detects functionally distinct coronavirus antibody features in children and elderly. *Nature communications*. 2021;12:2037.
- [22] Metsalu T, Vilo J. ClustVis: a web tool for visualizing clustering of multivariate data using Principal Component Analysis and heatmap. *Nucleic Acids Res*. 2015;43:W666-70.

Reviewers' Comments:

Reviewer #2:

Remarks to the Author:

Authors have meticulously addressed the reviewers comments. Particularly, I am pleased with the multitude of sensitivity testing performed by the authors to show the robustness of the observed results in their dataset. However, there are two issues I would like the authors to thoroughly address in their manuscript.

First, the authors main claim is that "For most genotypes, there was evidence of differential recognition of lineage-specific antigens and in some cases of a sufficient magnitude to suggest that some lineages should be considered antigenically distinct within their respective genotypes". Although these results seem robust, I believe there might be a significant geographical component in their dataset, which could be masking some of the results.

I would like the authors to assess this issue by taking their most representative variant-level variable dataset, which is the HPV16 dataset and do the following:

- 1) Select only the results derived from residents from the Americas and estimate the lineage-specific fold difference in neutralizing antibody titer for different variant lineages. Differential results between D lineage-specific sera compared to any other lineage-specific sera would be important to report as it would likely indicate human host genetic ancestry driven component in the results.
- 2) Select only the results derived from residents from Africa and estimate the lineage-specific fold difference in neutralizing antibody titer for different variant lineages. Differential results between B and C lineage-specific sera compared to A or D lineage-specific sera would be important to report.
- 3) Select only the results derived from residents from Europe and Asia and estimate the lineage-specific fold difference in neutralizing antibody titer for different variant lineages. Differential results between A lineage-specific sera compared to any other lineage-specific sera would be important to report.

Second, I would be pleased so see a discussion in the manuscript about the presented results on variant level antigenic variation, and the idea that antigenic variation plays a role in oncogenic HPVs evolutionary selective distribution in humans. That is, results presented here – particularly regarding HPV33 , HPV52 and HPV58 - are very interesting to reflect with the recent findings in a study by Pimenoff et al. (2023) Cell Host Microbe showing the long term effect of HPV vaccination leading to the likely replacement of vaccine targeted HPVs by the HPV33/52/58 types and that this could be due to antigenic variation playing a role in this selective evolutionary process of HPVs type (and variant) level replacement in time after vaccination.

We thank the reviewer for their time and consideration in reviewing our manuscript “Global Evaluation of Lineage-Specific Human Papillomavirus Capsid Antigenicity using Antibodies Elicited by Natural Infection”. Our responses to the comments raised are highlighted in blue below.

Reviewer #2 (Additional remarks to the author):

Authors have meticulously addressed the reviewers’ comments. Particularly, I am pleased with the multitude of sensitivity testing performed by the authors to show the robustness of the observed results in their dataset. However, there are two issues I would like the authors to thoroughly address in their manuscript.

First, the authors main claim is that “For most genotypes, there was evidence of differential recognition of lineage-specific antigens and in some cases of a sufficient magnitude to suggest that some lineages should be considered antigenically distinct within their respective genotypes”. Although these results seem robust, I believe there might be a significant geographical component in their dataset, which could be masking some of the results.

I would like the authors to assess this issue by taking their most representative variant-level variable dataset, which is the HPV16 dataset and do the following:

- 1) Select only the results derived from residents from the Americas and estimate the lineage-specific fold difference in neutralizing antibody titer for different variant lineages. Differential results between D lineage-specific sera compared to any other lineage-specific sera would be important to report as it would likely indicate human host genetic ancestry driven component in the results.
- 2) Select only the results derived from residents from Africa and estimate the lineage-specific fold difference in neutralizing antibody titer for different variant lineages. Differential results between B and C lineage-specific sera compared to A or D lineage-specific sera would be important to report.
- 3) Select only the results derived from residents from Europe and Asia and estimate the lineage-specific fold difference in neutralizing antibody titer for different variant lineages. Differential results between A lineage-specific sera compared to any other lineage-specific sera would be important to report.

We thank the reviewer for his/her positive comments and the opportunity to conduct further analyses of our dataset. We have conducted these evaluations and present the results in **Supplementary Figure 7**. The estimated antigenic distance between lineage A and C, for example, varies between 2 and 3-fold depending on which geographic region is represented. If samples from Africa are included the distance estimate is closer to 3, suggesting some geographical component to the antigenic distance. However, these estimates are likely to be influenced by both the ecological dispersal of lineages within each region and the representation of samples from each region included in this study. For instance, lineages B and C are uncommon outside of Africa (Clifford et al., 2019; Ref #10 in manuscript) and therefore not similarly represented by samples from The Americas or Asia. Thus, it is not possible to separate the influence of geography (and the possible impact of ancestry) from the ecological dispersal of lineages within that geographical location. We have added some text on this issue (lines 196-200, 318-320).

Second, I would be pleased so see a discussion in the manuscript about the presented results on variant level antigenic variation, and the idea that antigenic variation plays a role in oncogenic HPVs evolutionary selective distribution in humans. That is, results presented here – particularly regarding HPV33 , HPV52 and HPV58 - are very interesting to reflect with the recent findings in a study by Pimenoff et al. (2023) Cell Host Microbe showing the long term effect of HPV vaccination leading to the likely replacement of vaccine targeted HPVs by the

HPV33/52/58 types and that this could be due to antigenic variation playing a role in this selective evolutionary process of HPVs type (and variant) level replacement in time after vaccination.

We thank the reviewer for highlighting the publication by Pimenoff et al., 2023 (which was published after submission of this manuscript) and for the opportunity to cite this important study. We have previously published data demonstrating that at least for some types (e.g., HPV33, 45, 52 and 58) there are significant differences in serum neutralizing antibody sensitivity for some variants compared to the reference lineage (Refs. #31-36 in manuscript). These include reductions in sensitivity to cross-reactive antibodies elicited by the bivalent and quadrivalent vaccines, type-specific antibodies elicited by the nonavalent vaccine and neutralizing monoclonal antibodies. The present study examined lineage-specific natural infection antibodies and highlighted similar patterns of lineage-specific antibody specificity. That is, the same lineages are implicated in the differential sensitivity to natural infection and vaccine-derived antibodies (see lines 264-271). We have already speculated that some of these variant residues may have arisen or been maintained due to humoral immune selection (lines 279-286). It would be reasonable to speculate that the differential sensitivity of certain lineages to vaccine antibodies may impact the ecological diversity of HPV variants following introduction of a national vaccine programme, perhaps particularly where outlier antigenically distant variants are prevalent. Thus, the mechanism (s) underpinning apparent temporal changes to the ecological niche occupation of particular genotypes outlined in the report by Pimenoff et al., may well apply to lineage variants as well. We agree with the reviewer that this is an important and logical line of inquiry from this work and one that we are keenly aware of (see Refs #30-36 and particularly #35). However, at present this is purely speculative based upon *in vitro* data and observational estimates of lineage dispersal but something that in time can be evaluated empirically as vaccine programmes are rolled out globally and monitored through public health surveillance programmes. We have added some text to the discussion on this important topic area (see lines 350-352).

We agree that the revisions suggested have greatly improved the manuscript and we now resubmit this revised version for consideration for publication in the journal *Nature Communications*.

Yours sincerely and on behalf of the co-authors,

Simon Beddows, Ph.D.

Reviewers' Comments:

Reviewer #2:

Remarks to the Author:

I thank the authors for provided the additional analyses to take into account the geographical origin of the samples and the seropositivity rate differences in the dataset. Indeed, because the full dataset is biased with the geographical origin of the samples and there is a clear variation in the seropositivity rate, it is very interesting to see the geographically stratified analysis - the likely effect of differential genetic origin of the different HPV16 variant lineages. That is, when only European and Asian samples are analysed together for reasonable sample size, the antigenic distances are virtually non-significant (≤ 2) whereas when only African samples are analysed there is the strongest distance between the African origin HPV16 lineage C and the Eurasian origin lineage A (>3).

Although the sample size in stratified geographical sets is too small to make strong interpretations I would like the authors to note in the discussion this difference in antigenic distances between the HPV16 variants when using only African or using only European and Asian samples.

The authors should also do the same stratification for the HPV52 and HPV58 variant analysis (where the antigenic distances are the strongest with full dataset) and report if the same difference in the antigenic distances patterns is seen between African only and Eurasian only samples analysis between the A and the C variants.

We thank the reviewer for their time and consideration in reviewing our manuscript “Global Evaluation of Lineage-Specific Human Papillomavirus Capsid Antigenicity using Antibodies Elicited by Natural Infection”. Our responses to the comments raised are highlighted in blue below.

Reviewer #2 (Remarks to the author):

I thank the authors for provided the additional analyses to take into account the geographical origin of the samples and the seropositivity rate differences in the dataset. Indeed, because the full dataset is biased with the geographical origin of the samples and there is a clear variation in the seropositivity rate, it is very interesting to see the geographically stratified analysis - the likely effect of differential genetic origin of the different HPV16 variant lineages. That is, when only European and Asian samples are analysed together for reasonable sample size, the antigenic distances are virtually non-significant (≤ 2) whereas when only African samples are analysed there is the strongest distance between the African origin HPV16 lineage C and the Eurasian origin lineage A (>3). Although the sample size in stratified geographical sets is too small to make strong interpretations, I would like the authors to note in the discussion this difference in antigenic distances between the HPV16 variants when using only African or using only European and Asian samples.

We thank the reviewer for their interest in this study and welcome the additional comments. It has been an interesting endeavour to conduct additional analyses of these data to delineate the impact of geographical origin of the samples on the resulting antigenicity. We have added a line in the discussion to highlight the impact of geography on the antigenic distance between HPV16 lineages A and C (lines 320 – 323).

The authors should also do the same stratification for the HPV52 and HPV58 variant analysis (where the antigenic distances are the strongest with full dataset) and report if the same difference in the antigenic distances pattern is seen between African only and Eurasian only samples analysis between the A and the C variants.

In response to this comment by the reviewer we have attempted these additional analyses with HPV52 and HPV58. In all cases antigenic maps could be drawn and we present a summary of the outcomes below.

Type	Region	N	Fold	B	C	D
HPV52	Africa	A=13 B=0 C=0 D=0	A	3.0	1.2	3.2
			B		3.0	4.4
			C			3.7
	Eurasia (Asia and Europe)	A=3 B=16 C=7 D=3	A	2.5	1.3	3.5
			B		2.0	3.2
			C			3.1
HPV58	Africa	A=0 B=9 C=5 D=2	A	1.5	5.0	1.6
			B		5.5	1.7
			C			7.8
	Eurasia (Asia and Europe)	A=15 B=0 C=0 D=0	A	2.0	45.5	1.2
			B		41.8	1.8
			C			49.8

Unfortunately, in many cases, lineages were either represented by very few serum samples or by none. For example, for HPV52, only seropositive sera representing lineage A were available from Africa compared to a small number of samples representing all four lineages from Eurasia (Asian and European samples). This

was to be expected given the geographical and lineage distribution of samples collected (Supplementary Figure 1) and the rates of seropositivity (Supplementary Table 1). Thus, although lineage D remained the outlier lineage (compare with manuscript Figure 4 and Supplementary Figures 3, 6 and 8) the influence of geography on the estimated distances could not be reliably separated from the influence of the lack of representative samples. This is in contrast to the other analyses that we have performed on HPV16 (see Supplementary Figure 7) where the number of samples included was sufficient to answer the question appropriately. We have opted not to add these analyses to the Supplementary Information file as we believe that the estimated distances are unreliable and should not be afforded the same weight as the other antigenic distances that we have estimated in this study. We do, however, agree with the reviewer's premise and have expanded statements in the Results (lines 201 – 202) and Discussion (lines 325 - 326) to cover this.

We agree that the revisions suggested have greatly improved the manuscript and we now resubmit this revised version for consideration for publication in the journal *Nature Communications*.

Yours sincerely and on behalf of the co-authors,

Simon Beddows, Ph.D.